# Spectral Graph Coarsening Using Inner Product Preservation and the Grassmann Manifold

## Abstract

In this work, we propose a new functorial graph coarsening approach that preserves inner products between node features. Existing graph coarsening methods often overlook the mutual relationships between node features, focusing primarily on the graph structure. By treating node features as functions on the graph and preserving their inner products, our method ensures that the coarsened graph retains both structural and feature relationships, facilitating substantial benefits for downstream tasks. To this end, we present the Inner Product Error (IPE) that quantifies how well inner products between node features are preserved. By leveraging the underlying geometry of the problem on the Grassmann manifold, we formulate an optimization objective that minimizes the IPE, even for unseen smooth functions. We show that minimizing the IPE also promotes improvements in other standard coarsening metrics. We demonstrate the effectiveness of our method through visual examples that highlight its clustering ability. Additionally, empirical results on benchmarks for graph coarsening and node classification show superior performance compared to state-of-the-art methods.

## 1 Introduction

Graph-structured data has become ubiquitous in a wide range of domains, including social networks (Wasserman & Faust, 1994), biological systems (Pavlopoulos et al., 2011), and recommendation systems (Van Steen, 2010), due to its ability to model complex relationships and interactions. With the exponential increase in data availability, the size of graphs in many applications has also grown significantly. This surge in graph size presents major challenges, as traditional and even advanced graph processing techniques often become computationally infeasible or excessively time-consuming when applied to large-scale graphs. To address these issues, graph reduction techniques aim to simplify large graphs while retaining key structural features, thereby enhancing computational efficiency. There are three main strategies for graph reduction (Hashemi et al., 2024): graph sparsification, graph condensation, and graph coarsening.

Graph sparsification (Batson et al., 2009; Wickman et al., 2022) reduces graph size by selectively removing edges and nodes while maintaining overall structural properties. However, there is a limit to how much a graph can be sparsified without compromising its integrity. Graph condensation methods (Jin et al., 2021; Liu et al., 2023) aim to reduce graph size by generating a smaller, synthetic graph that replicates the performance of the original graph on specific tasks, such as training a Graph Neural Network (GNN). While condensation significantly lowers computational costs, it may not retain a clear structural interpretation, making it challenging to understand how or why certain nodes or edges are represented in the reduced version.

In contrast, graph coarsening (Chen et al., 2022) is a more traditional approach that reduces the size of a graph by grouping similar nodes into super-nodes, aiming to approximate the structure of the original graph. Coarsening methods aim to preserve essential structural properties such as the graph's spectral characteristics (Loukas & Vandergheynst, 2018), connectivity (LeFevre & Terzi, 2010), and community structure (Tian et al., 2008; Amiri et al., 2018). Most existing graph coarsening methods primarily focus on the structural properties of the graph and often overlook the

node features, which play a critical role in many graph learning tasks. These methods typically operate solely on the graph topology, neglecting the rich information encoded in the node attributes. Recently, the Featured Graph Coarsening (FGC) method was proposed to address this limitation by incorporating node features into the coarsening process (Kumar et al., 2023). FGC emphasizes the reconstruction of the node features after coarsening and promotes smoothness in the coarsened graph as part of its optimization objective. However, FGC does not fully exploit the relationships between different node attributes, which can encode valuable information about the graph's structure.

In this work, we propose a new approach to graph coarsening from a functorial perspective. We treat node features as functions, or signals, defined on the graph, focusing on preserving the relationships between these functions by maintaining their inner products during coarsening. Our method introduces a new coarsening metric, the Inner Product Error (IPE), which measures how well the inner products between graph signals are preserved. We postulate that minimizing IPE ensures the coarsened graph retains structural consistency and node feature relationships crucial for graph learning tasks. We exploit the geometry of the problem by recognizing that both the coarsening operator and the matrix spanning the node features (under a smoothness assumption) can be viewed as points on the Grassmann manifold. Leveraging the properties of the Grassmann manifold, we extend IPE minimization beyond observed node features, enabling our method to generalize to unseen features that satisfy the smoothness assumption. This approach is formulated as an optimization problem, and we compute the coarsening operator using gradient descent. Additionally, we theoretically show that minimizing our proposed approach leads to improvements in common graph coarsening metrics.

To demonstrate the effectiveness of our method, we present visual and empirical results showing that, while our method focuses on the functional relationships between node features, it also captures the global structure of the graph. We further validate its performance through extensive experiments on multiple graph coarsening and node classification benchmarks, where our method consistently outperforms state-of-the-art coarsening methods in terms of various established coarsening metrics, demonstrating its practical utility.

## 2 BACKGROUND

### 2.1 GRASSMAN MANIFOLD

The set of $n \times k$ matrices whose columns are orthonormal vectors forms a Riemannian manifold called the Stiefel manifold (James, 1976) defined by,

$$\mathrm{St}(n, k) := \{\boldsymbol{U} \in \mathbb{R}^{n \times k} | \boldsymbol{U}^T \boldsymbol{U} = \boldsymbol{I}_{k \times k}\}. \tag{1}$$

where $\boldsymbol{I}_{k \times k}$ is a rank-$k$ identity matrix.

The Grassmann manifold $\mathrm{Gr}(n, k)$ is a quotient manifold representing the set of $k$-dimensional subspaces of the Euclidean space $\mathbb{R}^n$. Two points on the Stiefel manifold that span the same subspace represent the same point on the Grassmann manifold (Bendokat et al., 2020; Edelman et al., 1998). In general, a point on $\mathrm{Gr}(n, k)$ is represented by an equivalence class

$$[\boldsymbol{U}] = \{\boldsymbol{U}\boldsymbol{O} : \boldsymbol{O} \in \mathcal{O}(k)\}, \tag{2}$$

where $\boldsymbol{U} \in \mathrm{St}(n, k)$ and $\mathcal{O}(k)$ is the group of $k \times k$ orthogonal matrices satisfying $\boldsymbol{O}^T \boldsymbol{O} = \boldsymbol{O}\boldsymbol{O}^T = \boldsymbol{I}$. The principal angles between two subspaces $\boldsymbol{U}_1$ and $\boldsymbol{U}_2$ are the angles that measure the smallest angular separation between basis vectors in one subspace and basis vectors in the other subspace. We denote them by $\boldsymbol{\theta} = [\theta^{(1)}, \theta^{(2)}, \ldots, \theta^{(k)}]$. Given two subspaces $\boldsymbol{U}_1$ and $\boldsymbol{U}_2$ on the Grassmann manifold $\mathrm{Gr}(n, k)$, the cosine of the principal angles between them can be computed using the SVD decomposition of $\boldsymbol{U}_1^T \boldsymbol{U}_2 = \boldsymbol{A}\boldsymbol{\Theta}\boldsymbol{B}^T$, where the singular values on the diagonal of $\boldsymbol{\Theta}$ are $[\cos(\theta^{(1)}), \cos(\theta^{(2)}), \ldots, \cos(\theta^{(k)})]$.

The geodesic similarity on the Grassmann manifold is defined using the principal angles between the two subspaces (Edelman et al., 1998). In Cohen & Talmon (2024) the authors showed that this geodesic similarity can be computed by,

$$G(\boldsymbol{U}_1, \boldsymbol{U}_2) = \sum_{i=1}^{k} \cos^2(\theta^{(i)}) = tr(\boldsymbol{U}_1 \boldsymbol{U}_1^T \boldsymbol{U}_2 \boldsymbol{U}_2^T). \tag{3}$$

## 2.2 GRAPH COARSENING

A graph with node features is denoted by the quadruplet $\mathcal{G} = (\mathcal{V}, \mathcal{E}, \boldsymbol{W}, \boldsymbol{X})$, where $\mathcal{V}$ is a set of $n$ vertices, $\mathcal{E}$ is a set of edges, $\boldsymbol{W} \in \mathbb{R}^{n \times n}$ is a weighted adjacency matrix, and $\boldsymbol{X} \in \mathbb{R}^{n \times p}$ is a node features matrix such that each row specifies the values of the $p$ features for each node. Each node feature, represented as a column of $\boldsymbol{X}$, can also be considered as a graph signal $\boldsymbol{x} \in \mathbb{R}^n$, assigning a real value to each vertex, namely, $\boldsymbol{x} : \mathcal{V} \to \mathbb{R}$. The graph Laplacian matrix $\boldsymbol{L}$ is defined by:

$$\boldsymbol{L} = \boldsymbol{D} - \boldsymbol{W}, \tag{4}$$

where $\boldsymbol{D} = \mathrm{diag}(\boldsymbol{W}\boldsymbol{1})$ is the diagonal degree matrix. The inner product between two graph signals $\boldsymbol{x}, \boldsymbol{y} \in \mathbb{R}^n$ with respect to the graph $\mathcal{G}$ is defined by:

$$\langle \boldsymbol{x}, \boldsymbol{y} \rangle_L = \boldsymbol{x}^T \boldsymbol{L} \boldsymbol{y} = \sum_{(i,j) \in E} w_{ij} (\boldsymbol{x}(i) - \boldsymbol{x}(j))(\boldsymbol{y}(i) - \boldsymbol{y}(j)) \tag{5}$$

where $w_{ij}$ are edge weights, and $\boldsymbol{x}(i), \boldsymbol{y}(i)$ are the values of the features at node $i$. We note that since $\boldsymbol{L}$ is a positive semi-definite matrix, $\boldsymbol{x}^\top \boldsymbol{L} \boldsymbol{x}$ induces a semi-norm and defines an inner product on the subspace of $\mathbb{R}^n$ orthogonal to the constant vector $\boldsymbol{1}$, as discussed in prior work Von Luxburg (2007). We denote the graph Laplacian eigenvalue decomposition by:

$$\boldsymbol{L} = \boldsymbol{U}\boldsymbol{\Lambda}\boldsymbol{U}^T, \tag{6}$$

where the columns of $\boldsymbol{U}$ are the eigenvectors of $\boldsymbol{L}$, and $\boldsymbol{\Lambda}$ is a diagonal matrix consisting of the corresponding eigenvalues.

Given a graph $\mathcal{G} = (\mathcal{V}, \mathcal{E}, \boldsymbol{W}, \boldsymbol{X})$ with $n$ nodes, the goal of graph coarsening is to construct a coarsened graph $\mathcal{G}_c = (\mathcal{V}_c, \mathcal{E}_c, \boldsymbol{W}_c, \boldsymbol{X}_c)$ with $k \ll n$ nodes, while preserving the main structural properties of $\mathcal{G}$, thereby simplifying subsequent analysis and computations. The coarsening procedure is defined through a linear mapping $\pi : \mathcal{V} \to \mathcal{V}_c$ that maps nodes in $\mathcal{G}$ to nodes in $\mathcal{G}_c$, termed 'super-nodes'. This linear mapping is defined by the coarsening matrix $\boldsymbol{P} \in \mathbb{R}_+^{k \times n}$, such that $\boldsymbol{X}_c = \boldsymbol{P}\boldsymbol{X}$. Each non-zero entry of $\boldsymbol{P}$ indicates a mapping from a node in $\mathcal{G}$ to a super-node in $\mathcal{G}_c$, i.e., if the $j$-th node in $\mathcal{G}$ is mapped to the $i$-th super-node of $\mathcal{G}_c$, then $\boldsymbol{P}_{i,j} > 0$. Let $\boldsymbol{L} \in \mathbb{R}^{n \times n}$ and $\boldsymbol{L}_c \in \mathbb{R}^{k \times k}$ be the respective Laplacian matrices of $\mathcal{G}$ and $\mathcal{G}_c$, and let $\boldsymbol{L}_l \in \mathbb{R}^{n \times n}$ and $\boldsymbol{X}_l \in \mathbb{R}^{n \times p}$ be the lifted Laplacian and feature matrices, i.e., the reconstructed full graph matrices after the coarsening procedure. The relationships between the coarse graph Laplacian and features and the original graph Laplacian and features are (Loukas, 2019):

$$\boldsymbol{L}_c = \boldsymbol{C}^\top \boldsymbol{L} \boldsymbol{C}, \qquad\qquad \boldsymbol{X}_c = \boldsymbol{P}\boldsymbol{X} \tag{7}$$

$$\boldsymbol{L}_l = \boldsymbol{P}^\top \boldsymbol{L}_c \boldsymbol{P}, \qquad\qquad \boldsymbol{X}_l = \boldsymbol{C}\boldsymbol{X}_c \tag{8}$$

where $\boldsymbol{C} \in \mathbb{R}_+^{n \times k}$ is the pseudo-inverse of $\boldsymbol{P}$, i.e., $\boldsymbol{C} = \boldsymbol{P}^\dagger$. The non-zero entries of $\boldsymbol{C}$ also imply a node mapping from $\mathcal{G}$ to $\mathcal{G}_c$, such that $\boldsymbol{C}_{i,j} > 0$ if the $i$-th node of $\mathcal{G}$ is mapped to the $j$-th super-node of $\mathcal{G}_c$. We note that the matrix $\boldsymbol{C}$ belongs to the following set:

$$\mathcal{C} = \{\boldsymbol{C} \geq 0 | \langle \boldsymbol{C}_{:,i}, \boldsymbol{C}_{:,j} \rangle = 0 \quad \forall i \neq j, \tag{9}$$
$$\langle \boldsymbol{C}_{:,i}, \boldsymbol{C}_{:,i} \rangle = d_i, \|\boldsymbol{C}_{:,i}\|_0 \geq 1, \|\boldsymbol{C}_{i,:}\|_0 = 1\}$$

where $\boldsymbol{C}_{:,i}$ and $\boldsymbol{C}_{:,j}$ are the $i$-th and $j$-th orthogonal columns of $\boldsymbol{C}$, $\boldsymbol{C}_{i,:}$ is the $i$-th row of $\boldsymbol{C}$, $\langle \cdot, \cdot \rangle$ is the standard inner product, and $d_i$ is some positive number. We note that, since the columns of a valid coarsening matrix $\boldsymbol{C}$ are orthogonal, it can also be viewed as a point on the Grassmann manifold $\mathrm{Gr}(n, k)$.

There are numerous evaluation metrics for graph coarsening methods, each tailored to consider different structural properties and suitable for specific applications. These metrics quantify the effectiveness of graph coarsening algorithms by assessing how the graph's main properties are preserved in the reduction process.

**Definition 1 (Relative Eigen Error (REE) (Loukas & Vandergheynst, 2018))** *The Relative Eigen Error (REE) is defined as REE* $= \frac{1}{k} \sum_{i=1}^k \frac{\lambda_{c,i} - \lambda_i}{\lambda_i}$, *where $\lambda_i$ and $\lambda_{c,i}$ are the $k$ dominant eigenvalues of the original graph Laplacian matrix $\boldsymbol{L}$ and the coarsen graph Laplacian matrix $\boldsymbol{L}_c$, respectively.*

**Definition 2 (Reconstruction Error (RE) (Liu et al., 2018))** *The Reconstruction Error (RE) between the original graph Laplacian $\boldsymbol{L}$ and the lifted graph Laplacian $\boldsymbol{L}_l$ is defined by, $RE = \|\boldsymbol{L} - \boldsymbol{L}_l\|_F^2$.*

The REE and RE are coarsening metrics independent of the graphs' node features. The REE measures the spectral similarity between graphs and how well global properties such as important edges are preserved, while the RE measure quantifies how much local information is preserved during the coarsening process.

**Definition 3 (Hyperbolic Error (HE) (Bravo Hermsdorff & Gunderson, 2019))** *The Hyperbolic Error (HE) between the original Laplacian matrix $\boldsymbol{L}$ and lifted Laplacian matrix $\boldsymbol{L}_l$ is defined as $HE = \text{arccosh}\left(1 + \frac{\|(\boldsymbol{L}-\boldsymbol{L}_l)\boldsymbol{X}\|_F^2 \|\boldsymbol{X}\|_F^2}{2tr(\boldsymbol{X}^T\boldsymbol{L}\boldsymbol{X})tr(\boldsymbol{X}^T\boldsymbol{L}_l\boldsymbol{X})}\right)$, where $\boldsymbol{X}$ is the node features matrix of the original graph.*

**Definition 4 (Dirichlet Energy Error (DEE))** *The Dirichlet Energy (DE) of a graph is defined by $DE = tr(\boldsymbol{X}^\top \boldsymbol{L}\boldsymbol{X})$, where $\boldsymbol{L}$ denotes the graph Laplacian and $\boldsymbol{X}$ denotes the node feature matrix of the graph (Kalofolias & Perraudin, 2017). We define the Dirichlet Energy Error (DEE) between the original graph $\mathcal{G}$ and its coarsened version $\mathcal{G}_c$ as $DEE = \left|\log\left(\frac{DE_\mathcal{G}}{DE_{\mathcal{G}_c}}\right)\right|$, where $DE_\mathcal{G}$ and $DE_{\mathcal{G}_c}$ are the DE of the original and coarsened graphs, respectively.*

The HE and DEE are coarsening measures that also consider the node features. The HE measures the distortion in the geometric structure of the data with respect to the hyperbolic space. This is particularly useful when the graph has a hierarchical structure (e.g., trees), as hyperbolic spaces are well-suited for representing such data. The DE measures the smoothness of the node features on a graph; lower DE values suggest that the node features are closely aligned with the graph structure. Consequently, we define the DEE which quantifies the extent to which the intrinsic graph structure in the node features is preserved during the coarsening process. We note that the authors in Kumar et al. (2023) suggest minimizing the DE of the coarsened graph as part of their graph coarsening optimization objective.

## 3 PROPOSED METHOD

---

**Algorithm 1** INGC Algorithm

**Input**: $\boldsymbol{L} \in \mathbb{R}^{n \times n}, \boldsymbol{X} \in \mathbb{R}^{n \times p}$
**Parameters**: $\beta, \lambda, \alpha, \eta, t_{iter}, c_{iter}$
**Output**: $\boldsymbol{L}_c \in \mathbb{R}^{k \times k}, \boldsymbol{X}_c \in \mathbb{R}^{k \times p}$
1: Compute $\boldsymbol{U}^{(k)}$, the $k$-leading eigenvectors of $\boldsymbol{L}$.
2: Initialize $\boldsymbol{C}_0, t = 0$.
3: **while** $\|\boldsymbol{C}_{t+1} - \boldsymbol{C}_t\|_F < \epsilon_C$ or $t < t_{iter}$ **do**
4:     $\boldsymbol{C}_{t(0)} = \boldsymbol{C}_t$.
5:     Compute $\nabla_C f(\boldsymbol{X}_c, \boldsymbol{C})$ according to equation 18
6:     **for** $i$ in range($c_{iter}$) **do**
7:         Update $\boldsymbol{C}_{t+1}$ using the gradient descent step:
    $\boldsymbol{C}_{t(i+1)} \leftarrow \boldsymbol{C}_{t(i)} - \eta \nabla_C f(\boldsymbol{X}_{c(t)}, \boldsymbol{C}_{t(i)})$
8:     **end for**
9:     $\boldsymbol{C}_{t+1} = \boldsymbol{C}_{t(c_{iter})}, \boldsymbol{X}_{c(t+1)} = \boldsymbol{C}_{t+1}^\dagger X$
10:     $t = t + 1$
11: **end while**
12: **return** $\boldsymbol{L}_c = \boldsymbol{C}_t^\top \boldsymbol{L}\boldsymbol{C}, \boldsymbol{X}_c = \boldsymbol{C}_t^\dagger X$

**Algorithm 2** SINGC Algorithm

**Input**: $\boldsymbol{L} \in \mathbb{R}^{n \times n}$
**Parameters**: $\lambda, \alpha, \eta, t_{iter}$
**Output**: $\boldsymbol{L}_c \in \mathbb{R}^{k \times k}, \boldsymbol{X}_c \in \mathbb{R}^{k \times p}$
1: Compute $\boldsymbol{U}^{(k)}$, the $k$-leading eigenvectors of $\boldsymbol{L}$.
2: Initialize $\boldsymbol{C}_0, t = 0$.
3: **while** $\|\boldsymbol{C}_{t+1} - \boldsymbol{C}_t\|_F < \epsilon_C$ or $t < t_{iter}$ **do**
4:     Compute $\nabla_C f(\boldsymbol{C})$ according to equation 19
5:     Update $\boldsymbol{C}_{t+1}$ using the gradient descent step:
    $\boldsymbol{C}_{t+1} \leftarrow \boldsymbol{C}_t - \eta \nabla_C f(\boldsymbol{C}_t)$
6:     $t = t + 1$
7: **end while**
8: **return** $\boldsymbol{L}_c = \boldsymbol{C}_t^\top \boldsymbol{L}\boldsymbol{C}$

---

Our method adopts a functorial perspective for graph coarsening, focusing on maintaining the relationships among different functions defined on the graph throughout the coarsening process.

Specifically, it aims to preserve the inner products between various functions defined on the graph. To achieve this, we first introduce the following new graph coarsening metric that quantifies how well the inner products between all given graph signals (i.e., node features) are preserved during the coarsening process.

**Definition 5 (Inner Product Error (IPE))** *Let $L$ and $X$ be the original graph Laplacian and node features matrix, and let $L_c$ and $X_c$ be their respective coarsened graph Laplacian and features matrix. The Inner Product Error (IPE) is defined by IPE $= \|X^\top L X - X_c^\top L_c X_c\|_F^2$.*

The motivation for this approach is based on the following result.

**Theorem 1** *Let $L$ and $L_c$ be graph Laplacians of a graph $\mathcal{G}$ with $(n - k)$ connected components and its coarsened graph $\mathcal{G}_c$, respectively. If for any two graph signals $x, y \in \mathbb{R}^n$, the inner product between the two signals is preserved under the coarsening process, i.e.:*

$$x^\top L y = x_c^\top L_c y_c,$$

*then the graph Laplacian of the original graph, $L$, can be fully reconstructed from $L_c$ via:*

$$L = L_l = P^\top L_c P$$

See Appendix A.1 for the proof.

We note that in most practical cases Theorem 1 is not feasible, since a necessary condition for such a reconstruction is that the rank of $L$ is less than $k$. This implies that the graph has $(n - k)$ connected components – a condition that is rarely met. However, this theorem implies that by preserving the inner product of graph signals under the coarsening process, we also preserve the graph's key structural properties.

### 3.1 INNER PRODUCT PRESERVING GRAPH COARSENING

We propose the following optimization for graph coarsening that aims to preserve the inner products between graph signals. The objective function and constraints are formally defined as follows:

$$\min_C f(C) = \|X^\top L X - X_c^\top C^\top L C X_c\|_F^2 - \beta tr(U^{(k)}(U^{(k)})^\top C C^\top) + \lambda g(C) + \alpha h(L_c)$$

$$\text{s.t.} \quad L_c = C^T L C, X_c = C^\dagger X, C \in \mathcal{C} \tag{10}$$

where $C = P^\dagger \in \mathbb{R}^{n \times k}$ is the target coarsening operator; $L \in \mathbb{R}^{n \times n}$ and $X \in \mathbb{R}^{n \times p}$ are the given graph Laplacian and feature matrix of the original graph; $U^{(k)}$ is a matrix containing the $k$ leading eigenvectors of $L$; $L_c \in \mathbb{R}^{k \times k}$ and $X_c \in \mathbb{R}^{k \times p}$ are the Laplacian and feature matrix of the coarsened graph, and $\mathcal{C}$ is the set defined in equation 9. Functions $h(\cdot)$ and $g(\cdot)$ are regularization functions for $L_c$ and $C$, while $\lambda, \alpha > 0$ are positive regularization parameters.

The primary objective in Equation 10 is to minimize the IPE in the graph coarsening process. We achieve this using two complementary terms. The first term involves directly minimizing the IPE on the given node features. While this enhances performance on the available data, it does not generalize well to new signals, as its effectiveness depends heavily on the specific information encoded in the feature matrix $X$. The second term aims to minimize the IPE for general unseen signals that satisfy a smoothness assumption. This is accomplished by maximizing the alignment of the coarsening matrix $C$ with the leading eigenvectors $U^{(k)}$ of $L$, thereby maximizing their Grassmann similarity. In Section 3.3 we show that this alignment encourages the preservation of important structural properties of the graph as well as the inner product between unseen smooth signals. The parameter $\beta$ balances the two terms, adjusting the emphasis between performance on the observed data and generalization to new signals. We note that our empirical results show that both terms contribute significantly to the coarsening process.

In addition to the minimization of the IPE, we use two specific regularization functions, which were considered in Kumar et al. (2023), to promote both a balanced distribution across super-nodes and connectivity of the coarsened graph. Specifically, we use $g(C) = \|C^\top\|_{1,2}^2 = \sum_{i=1}^n \left( \sum_{j=1}^k |C_{i,j}| \right)^2$, an $l_{1,2}$-based group penalty, that as shown in (Ming et al., 2019; Kumar

et al., 2023) promotes a valid coarsening operator $C$. The second regularization is $h(L_c) = \log \det(L_c + J)$, where $J = \frac{1}{k}\mathbf{1}_{k \times k}$. This regularization ensures that $L_c + J$ is full rank, implying that $L_c$ has rank $k - 1$, which guarantees that the coarsened graph $\mathcal{G}_c$ is connected (Chung, 1997; Kalofolias, 2016).

### 3.2 PROPOSED ALGORITHMS

One limitation of the objective in Equation 10 is that the derivative of the first term does not have a closed-form expression with respect to $C$. As a remedy, we adopt the multi-block optimization framework suggested by Kumar et al. (2023), recasting our objective function as:

$$\min_{C} f(X_c, C) = \|X^\top L X - X_c^\top C^\top L C X_c\|_F^2 - \beta tr(U^{(k)}(U^{(k)})^\top C C^\top) \tag{11}$$

$$+ \lambda\|C^T\|_{1,2}^2 - \alpha \log \det(L_c + J)$$

$$\textbf{s.t.} \qquad X_c = C^\dagger X, \quad L_c = C^\top L C, C \in \mathcal{C}$$

In this recast, the modified objective function is differentiable with respect to $C$, and the gradients are presented in Appendix A.4. We optimize this objective by applying projected gradient descent (Bertsekas, 1997) to estimate the matrix $C$. A full description of this method is in Algorithm 1, termed INGC. Algorithm 2 is a simpler version, where we omit the first term in Equation 11. This simplification makes the optimization problem computationally more efficient and doesn't depend on the feature matrix $X$, and the matrix $C$ can be estimated using standard gradient descent. We refer to this algorithm as SINGC.

### 3.3 THEORETICAL ANALYSIS

Next, we present two key analytical results. First, we show that the second term in Equation 10 minimizes the IPE for smooth graph signals. Second, we establish the connection between this minimization, and common graph coarsening metrics, such as DEE and REE. We begin by defining what constitutes a smooth graph signal. The authors in Dong et al. (2016) model a smooth graph signal generation mechanism as:

$$x = Uh + \epsilon_\eta \eta, \tag{12}$$

where $L = U\Lambda U^\top$ is the Laplacian of the respective graph signal, $h \sim \mathcal{N}(0, \Lambda^\dagger) \in \mathbb{R}^n$, $\eta \sim \mathcal{N}(0, I_{n \times n}) \in \mathbb{R}^n$, and $\epsilon_\eta > 0$ is the noise standard deviation. This model suggests that a smooth graph signal is a combination of the first eigenvectors of $L$ and scaled noise.

**Assumption 1 ($k$-smooth graph signal (Dietrich et al., 2022))** *A graph signal $x \in \mathbb{R}^n$ is termed "$k$-smooth" on the graph $\mathcal{G}$ if it can be fully expressed by the first $k$ eigenvectors of its corresponding Laplacian $L$, i.e., $x = \sum_{i=1}^{k} c_i u^{(i)} = c^\top U^{(k)}$ where the columns of $U^{(k)} = [u^{(1)}, \dots, u^{(k)}] \in \mathbb{R}^{n \times k}$ are the $k$-leading eigenvectors of $L$.*

Building on Assumption 1, the following result provides the motivation for incorporating the Grassmann similarity score into our proposed objective.

**Theorem 2** *Let $X$ be a feature matrix of a graph $\mathcal{G}$ with a Laplacian matrix $L$, where each column of $X$ is $k$-smooth on the graph $\mathcal{G}$. Then, any mapping $L_c = CLC^\top, X_c = C^\top X$, such that $C = U^{(k)}O$ satisfies:*

$$\|X^\top L X - X_c^\top L_c X_c\|_F^2 = 0$$

*where the columns of $U^{(k)} \in \mathbb{R}^{n \times k}$ are the $k$-leading eigenvectors of $L$, and $O \in \mathcal{O}(k)$ is some $k$-dimension rotation matrix.*

See Appendix A.2 for proof. Theorem 2 implies that any coarsening operator $C$ whose columns span the same subspace as $U^{(k)}$ minimizes the IPE in Equation 5 for any $k$-smooth signals on the original graph. Thus, the second term in our objective in Equation 10 maximizes the Grassmann similarity between $C$ and $U^{(k)}$, aiming to find a valid coarsening operator (i.e., $C \in \mathcal{C}$) that satisfies $C = U^{(k)}O$. Next, we present a theorem that provides bounds on the DEE (4) and REE (1) as functions of $\epsilon$, which quantifies the deviation of the second term in our objective from its optimal value (if $C$ and $U^{(k)}$ span the same subspace, then $\text{tr}(U^{(k)}(U^{(k)})^\top C C^\top) = k$).

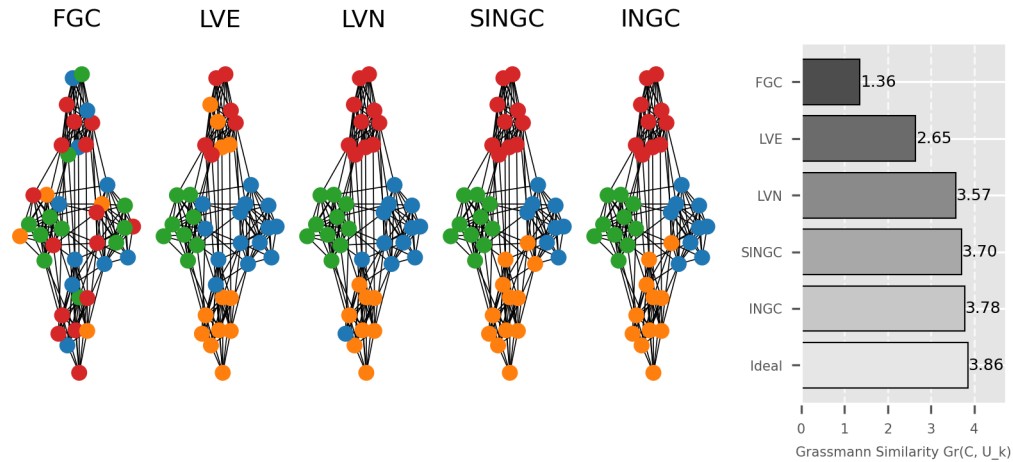

Figure 1: Nodes assignment of all methods on a synthetic graph generated from a Stochastic Block Model (SBM). Nodes with the same color belong to the same class (super-node). On the right, a bar plot presents the Grassmann similarity between the coarsening matrix $C$ obtained by each method and the leading four eigenvectors $U^{(k)}$ of the graph Laplacian. The bottom bar corresponds to the Grassmann similarity for the ideal partitioning, that is, between the coarsening matrix $C$ corresponding to the partitioning based on the graph's underlying block model and $U^{(k)}$. Note that the maximum Grassmann similarity in this case is 4.

**Theorem 3** *Let $L$ be the Laplacian of a connected graph $\mathcal{G}$, and let $U^{(k)}$ be the matrix containing its $k$-leading eigenvectors. Suppose $L_c = C^\top L C$ is the Laplacian of a coarsened graph derived using a coarsening operator $C$ such that $tr(U^{(k)}(U^{(k)})^\top C C^\top) = k - \epsilon$, and the columns of $C$ span the constant vector in $\mathbb{R}^n$. Then, the eigenvalues of the original graphs $\{\lambda^i\}_{i=1}^k$ and the coarsened graph $\{\lambda_c^i\}_{i=1}^k$, along with the Dirichlet energies of a $k$-smooth graph signal $x$ on the graph, satisfy the inequalities:*

$$(1 - \epsilon\kappa)^2 \|x\|_L \leq \|x_c\|_{L_c} \leq (1 + \epsilon\kappa)^2 \|x\|_L,$$

$$\frac{1}{\mu_1}\lambda^{(i)} \leq \lambda_c^{(i)} \leq \frac{1}{\mu_2}\frac{(1 + \epsilon\kappa)^2}{1 - (\epsilon\kappa)^2(\lambda^{(i)}/\lambda^{(2)})}\lambda^{(i)}, \qquad 2 \leq i \leq k$$

*whenever $\epsilon\kappa < \frac{\lambda^{(2)}}{\lambda^{(i)}}$. Here $\kappa = \frac{\lambda_{max}(L)}{\lambda^{(2)}}$, $\lambda_{max}(L)$ is the maximum eigenvalue of $L$, $P = C^\dagger$, $\mu_1, \mu_2$ and the first and $k$ eigenvalues of the matrix $PP^\top$, and $\|x\|_L = x^T L x$ and $\|x_c\|_{L_c} = x_c^T L_c x_c$.*

See Appendix A.3 for proof. Theorem 3 establishes that the DEE is bounded between $2\log(1 + \epsilon\kappa)$ and $2\log(1 - \epsilon\kappa)$, with additional bounds on some eigenvalues. This result clarifies the relationship between our objective and common graph coarsening metrics, showing that as the objective approaches its optimal value, these metrics yield improved results.

## 4 EXPERIMENTAL RESULTS

### 4.1 VISUAL ILLUSTRATION

A key aspect of graph coarsening is how well the global structure of the graph is preserved. This can be assessed by how effectively the partitioning into super-nodes captures the original graph's structure. To illustrate this property, we provide a visual example showing how the super-nodes generated by our method align with the graph's global structure. This example highlights how our approach maintains a meaningful graph representation, despite focusing solely on functional relationships in the coarsening objective.

The example is based on a synthetic graph generated using a Stochastic Block Model (SBM) (Abbe, 2017) with four classes, an intra-class probability of $p = 0.9$, and an inter-class probability

| Method | | Karate Club | | | Les Miserables | | | Cora | | | Citeseer | | | #Best | #2-Best |
|---|---|---|---|---|---|---|---|---|---|---|---|---|---|---|---|
| | $r$ | 0.7 | 0.5 | 0.3 | 0.7 | 0.5 | 0.3 | 0.7 | 0.5 | 0.3 | 0.7 | 0.5 | 0.3 | | |
| LVN | REE | **0.30** | 1.52 | 3.15 | 0.36 | 1.39 | 7.82 | **0.57** | 1.31 | 4.23 | **0.68** | 1.58 | 4.11 | 3 | 4 |
| | RE | 9.71 | 9.87 | 10.31 | 11.42 | 11.91 | 11.91 | 11.42 | 11.62 | 11.70 | 10.85 | 11.05 | 11.14 | 0 | 0 |
| | HE | 1.74 | 1.89 | 2.25 | 1.92 | 2.60 | 2.54 | 1.89 | 2.50 | 3.17 | 1.93 | 2.52 | 3.43 | 0 | 0 |
| | DEE | 36.2 | 47.8 | 46.6 | 65.1 | 99.4 | 90.2 | 39.1 | 70.7 | 90.2 | 40.0 | 66.3 | 111.1 | 0 | 0 |
| | IPE | 0.71 | 0.72 | 1.09 | 2.02 | 2.56 | 1.91 | 2.87 | 2.23 | 1.73 | 0.93 | 0.98 | 1.09 | 0 | 0 |
| LVE | REE | 0.82 | **0.48** | **2.26** | 1.05 | 4.56 | **7.60** | 0.81 | 1.94 | 5.14 | 0.79 | 1.62 | 4.26 | 3 | 2 |
| | RE | 9.39 | 9.91 | 9.97 | 11.55 | 12.14 | 12.48 | 10.62 | 11.51 | 11.69 | 10.52 | 11.02 | 11.14 | 0 | 0 |
| | HE | 1.40 | 1.93 | 2.31 | 1.63 | 2.16 | 2.89 | 1.29 | 2.31 | 3.10 | 1.61 | 2.50 | 3.40 | 0 | 0 |
| | DEE | 17.3 | 39.0 | 84.8 | 19.1 | 20.6 | 63.6 | 22.5 | 44.9 | 78.1 | 30.1 | 63.5 | 109.0 | 0 | 0 |
| | IPE | 0.66 | 0.88 | 0.92 | 1.59 | 2.77 | 3.87 | 0.58 | 1.19 | 1.54 | 0.61 | 0.79 | 0.88 | 0 | 0 |
| FGC | REE | 1.35 | 3.94 | 7.54 | 3.08 | 10.31 | 33.18 | 1.78 | 5.40 | 15.99 | 1.58 | 8.92 | 35.88 | 0 | 0 |
| | RE | 8.70 | 8.81 | 9.26 | 9.97 | 9.98 | 10.91 | 9.70 | 10.82 | 10.76 | 10.47 | 10.23 | 10.31 | 0 | 1 |
| | HE | 1.03 | 1.23 | 1.80 | 0.82 | 0.87 | 1.58 | 0.76 | 1.40 | 1.56 | 1.89 | 1.39 | 1.61 | 0 | 2 |
| | DEE | 8.05 | 11.57 | 21.36 | 4.78 | 7.65 | 9.74 | 0.20 | 0.41 | 5.01 | 19.5 | 7.87 | 1.34 | 0 | 1 |
| | IPE | 0.55 | 0.58 | 0.94 | 1.05 | 2.17 | 3.67 | 0.41 | 1.01 | 3.67 | 1.10 | 0.49 | 0.68 | 0 | 0 |
| INGC (Ours) | REE | 0.78 | 1.30 | 2.97 | **0.08** | **1.30** | 10.40 | 0.86 | **0.84** | **0.76** | 0.71 | **0.62** | **0.42** | 6 | 4 |
| | RE | 6.27 | **7.00** | **8.28** | **5.43** | **8.08** | **9.79** | **9.49** | **10.17** | **10.42** | **8.86** | 9.85 | 10.10 | 9 | 3 |
| | HE | 0.29 | **0.45** | **1.00** | **0.08** | **0.33** | **0.83** | **0.67** | **1.04** | **1.37** | **0.64** | 1.21 | 1.61 | 9 | 3 |
| | DEE | **0.01** | **0.03** | **0.02** | **0.04** | **0.02** | **0.10** | **0.03** | **0.40** | **3.12** | **0.02** | **0.66** | 1.01 | 11 | 1 |
| | IPE | 0.19 | **0.31** | **0.48** | 0.30 | **0.57** | **0.86** | **0.31** | **0.43** | **0.68** | **0.24** | 0.34 | 0.58 | 8 | 4 |
| SINGC (Ours) | REE | 0.86 | 1.78 | 4.02 | 0.50 | 2.32 | 7.78 | 0.86 | **0.84** | 5.10 | 0.83 | **0.62** | **0.42** | 3 | 0 |
| | RE | **6.09** | 8.05 | 8.74 | 9.07 | 9.60 | 10.49 | 9.54 | **10.17** | 10.95 | 9.32 | **9.76** | **9.92** | 4 | 7 |
| | HE | **0.27** | 0.85 | 1.46 | 0.51 | 0.72 | 1.29 | 0.69 | **1.04** | 1.84 | 0.83 | **1.14** | **1.27** | 4 | 7 |
| | DEE | 0.02 | 0.51 | 6.56 | 0.47 | 0.35 | **0.10** | **0.03** | **0.40** | 8.12 | 1.01 | 1.45 | **0.10** | 4 | 7 |
| | IPE | **0.19** | 0.43 | 0.59 | **0.21** | 0.63 | 1.07 | **0.31** | **0.43** | 0.69 | 0.27 | **0.25** | **0.47** | 6 | 6 |

Table 1: Comparison of coarsening methods on various datasets using different metrics and coarsening ratios ($r$). For each method we report the REE, RE, HE, DEE, INP at different coarsening ratios for the datasets Karate Club, Les Miserables, Cora, and Citeseer. The best performance for each metric is highlighted in bold, and the second-best is underlined. The last two columns indicate the number of times each method achieved the best and second-best performance across all settings.

of $q = 0.05$. The feature matrix $X$ is generated following the same graph signal generation mechanism described in Section 3.3. For this illustration, we set the target coarsened graph for all coarsening methods to have $k = 4$ super-nodes. In Figure 1, we present the results obtained by our methods (INGC/SINGC), alongside three other graph coarsening baselines: Feature-based Graph Coarsening (FGC) (Kumar et al., 2023), which incorporates node features into the coarsening process, and the Local Variation Neighborhood (LVN) and Local Variation Edges (LVE) methods (Loukas & Vandergheynst, 2018), which use the original graph Laplacian eigenvectors as part of their coarsening objectives. We observe that the assignment of the nodes to super-nodes stemming from our methods closely aligns with the partitioning of the nodes to four classes according to the SBM, as indicated by the node colors in Figure 1. On the right-hand side of Figure 1, we present a bar plot comparing the Grassmann similarity between the coarsening matrices $C$ (which encode the vertex partitioning) produced by each method and the top four eigenvectors of the original graph Laplacian, $U^{(k)}$. The bottom bar represents the matrix $C$ that encodes the ideal partitioning based on the graph's underlying block model, serving as a baseline. We observe that our method achieves the highest similarity, closely approaching the ideal partitioning. We note that when the coarsening matrix $C$ and the leading eigenvector matrix $U^{(k)}$ span the same subspace, the Grassmann similarity reaches its maximum of 4. This comparison highlights our motivation for incorporating Grassmann similarity into our coarsening objective, as it promotes preservation of the graph's global structure. Numerically, this property is reflected in the REE metric; we present an extensive evaluation of this and other metrics in the following section. Additional visual comparisons demonstrating practical coarsening scenarios are presented in Appendix B.

| Dataset | r | GCOND | SCAL(LV) | FGC | INGC(Ours) | SINGC(Ours) |
|---------|------|-------|----------|-----|------------|-------------|
| Cora | 0.3 | $81.56 \pm 0.6$ | $79.42 \pm 1.71$ | $\underline{85.79 \pm 0.24}$ | $\mathbf{87.55 \pm 0.16}$ | $84.51 \pm 0.33$ |
|      | 0.1 | $81.37 \pm 0.4$ | $71.38 \pm 3.62$ | $81.46 \pm 0.79$ | $\mathbf{83.38 \pm 0.47}$ | $\underline{82.76 \pm 0.32}$ |
|      | 0.05 | $\underline{79.93 \pm 0.44}$ | $55.32 \pm 7.03$ | $\mathbf{80.01 \pm 0.51}$ | $77.42 \pm 0.78$ | $77.81 \pm 0.68$ |
| Citeseer | 0.3 | $72.43 \pm 0.94$ | $68.87 \pm 1.37$ | $74.64 \pm 1.37$ | $\mathbf{76.89 \pm 0.23}$ | $\underline{76.66 \pm 0.27}$ |
|          | 0.1 | $70.46 \pm 0.47$ | $71.38 \pm 3.62$ | $\mathbf{73.36 \pm 0.53}$ | $\underline{72.63 \pm 0.25}$ | $69.71 \pm 0.72$ |
|          | 0.05 | $64.03 \pm 2.4$ | $55.32 \pm 7.03$ | $\mathbf{71.02 \pm 0.96}$ | $66.02 \pm 0.32$ | $66.37 \pm 0.57$ |
| Co-phy | 0.05 | $93.05 \pm 0.26$ | $73.09 \pm 7.41$ | $\underline{94.27 \pm 0.25}$ | $\mathbf{94.29 \pm 0.10}$ | $94.04 \pm 0.06$ |
|        | 0.03 | $92.81 \pm 0.31$ | $63.65 \pm 9.65$ | $\underline{94.02 \pm 0.20}$ | $\mathbf{94.20 \pm 0.13}$ | $93.52 \pm 0.13$ |
|        | 0.01 | $92.79 \pm 0.4$ | $31.08 \pm 2.65$ | $93.08 \pm 0.22$ | $\mathbf{93.95 \pm 0.20}$ | $\underline{93.20 \pm 0.10}$ |
| Pubmed | 0.05 | $78.16 \pm 0.3$ | $72.82 \pm 2.62$ | $80.73 \pm 0.44$ | $\mathbf{83.59 \pm 0.22}$ | $\underline{83.55 \pm 0.32}$ |
|        | 0.03 | $78.04 \pm 0.47$ | $70.24 \pm 2.63$ | $79.91 \pm 0.30$ | $\underline{81.93 \pm 0.22}$ | $\mathbf{83.19 \pm 0.18}$ |
|        | 0.01 | $77.2 \pm 0.20$ | $54.49 \pm 10.5$ | $78.42 \pm 0.43$ | $\underline{79.09 \pm 0.26}$ | $\mathbf{79.96 \pm 0.34}$ |
| Co-CS | 0.05 | $86.29 \pm 0.63$ | $34.45 \pm 10.0$ | $89.60 \pm 0.39$ | $\underline{90.84 \pm 0.12}$ | $\mathbf{90.92 \pm 0.22}$ |
|       | 0.03 | $86.32 \pm 0.45$ | $26.06 \pm 9.29$ | $88.29 \pm 0.79$ | $\underline{89.59 \pm 0.38}$ | $\mathbf{89.99 \pm 0.41}$ |
|       | 0.01 | $84.01 \pm 0.02$ | $14.42 \pm 8.5$ | $\underline{86.37 \pm 1.36}$ | $\mathbf{87.93 \pm 0.33}$ | $83.39 \pm 0.33$ |
| #Best | | 0 | 0 | 3 | 8 | 4 |
| #2-Best | | 1 | 0 | 4 | 5 | 5 |

Table 2: Node classification accuracy on various datasets for different coarsening ratios $r$ using various coarsening methods. Best results are in bold; second-best results are underlined. The last two rows indicate the number of times each method achieved the best and second-best performance.

## 4.2 GRAPH COARSENING METRICS

Here, we evaluate the performance of our methods, INGC and SINGC, on several benchmark datasets and compare them with other existing graph SOTA coarsening methods: LVN and LVE for preserving graph structural properties (e.g., REE, RE) and FGC for metrics incorporating node features. The evaluation is based on the coarsening metrics presented in Sec. 2.2 that assess different complementary aspects. We conduct experiments on four datasets: The Karate Club(Zachary, 1977), Les Miserables(Knuth, 1993), Cora(McCallum et al., 2000), and (Giles et al., 1998). Note that the Cora and Citeseer datasets include node features, whereas the Karate Club and Les Miserables datasets do not. For the latter two, we generated node features using the signal generation mechanism presented in Section 3.3.

Table 1 summarizes the performance of our methods (INGC and SINGC) and other baselines (FGC, LVN and LVE) across different datasets and coarsening ratios ($r = \frac{k}{n} = 0.7, 0.5$, and 0.3). The best performance for each metric is highlighted in bold, and the second-best is underlined. The last two columns summarize the number of settings in which each method achieved the lowest or second-lowest score compared to the other methods. We observe that our INGC method achieves the best overall performance across all graph metrics. SINGC, the simplified version of our method, also performs competitively, often demonstrating the second-best results in these metrics and, in some scenarios, even achieving the best performance. Our method's superior RE score demonstrates its broader applicability in preserving graph structure during coarsening, extending beyond Theorem 1 theoretical scenario, as all datasets used are connected or have far fewer than $n - k$ connected components. The baseline methods, LVN, LVE, and FGC, show varying performance on different metrics. The LVN and LVE show better performance on REE in certain datasets, indicating good preservation of spectral and global properties in these cases. However, it generally falls short in other metrics, particularly those involving node features. The FGC is the only baseline method that considers the node features as part of the coarsening process, and indeed it shows better performance than other baselines considering coarsening metrics that involve node features. We observe that both our methods outperform the FGC which is considered the SOTA method across many settings.

An important note is that the best performance w.r.t. each metric is achieved using different hyperparameters, which are reported in Appendix F.2. This variability underscores the importance of selecting hyperparameters based on a specific application and the coarsening metric most relevant to it. For example, minimizing a REE might be crucial for clustering applications. Conversely, applications such as graph pooling and node classification, which depend heavily on the node

features, would benefit from prioritizing DEE and INP in the hyperparameter tuning process. Our methods offer flexibility in this regard. By adjusting the hyperparameters $\beta$, $\lambda$, and $\alpha$ in the objective function (Equation 10), we can tailor the coarsening process to prioritize specific properties.

### 4.3 NODE CLASSIFICATION

We evaluate the graph coarsening by applying the coarsening methods to the task of node classification using several benchmark datasets. Node classification is a widely used benchmark for evaluating the efficacy of graph coarsening algorithms, as it tests a coarsened graph's ability to preserve essential structural and feature information necessary for accurate label prediction. In this experiment, a Graph Neural Network (GNN) is trained on the coarsened graph and employed to predict node labels for the original, full-sized graph. This approach speeds up GNN training time, as the coarsened graph has significantly fewer nodes and edges.

We follow the evaluation procedure described in Kumar et al. (2023), performing the following steps. First, we learn a coarsened graph from the original graph using the selected coarsening algorithm. Second, we compute labels for the super-nodes in the coarsened graph based on the learned coarsening operator $\boldsymbol{P} = \boldsymbol{C}^{\dagger}$, using $\boldsymbol{y}_c = \boldsymbol{P}\boldsymbol{y}$, where $\boldsymbol{y}$ represents the labels of the original graph. Third, we train a node classification GCN on the coarsened graph using these super-node labels. Finally, we evaluate the classification performance on the original graph by comparing the labels given by $\hat{\boldsymbol{y}} = \text{GCN}(L, X)$ to the full graph node labels ($\boldsymbol{y}$). It is important to note that this node classification task is performed solely for evaluation purposes, as we have access to all the labels $\boldsymbol{y}$ during the process.

In our experiments, we employ a Graph Convolutional Network (GCN) (Kipf & Welling, 2016) and compare the performance of our methods (INGC and SINGC) against current SOTA graph coarsening methods for node classification, including SCAL (Huang et al., 2021), Featured Graph Coarsening (FGC) (Kumar et al., 2023), and Graph Condensation (GCOND) (Jin et al., 2021). The datasets used in this experiment include two medium-sized graphs—Cora and Citeseer—and three large-scale graphs—Co-Physics, Pubmed, and Co-CS. The effectiveness of each method is assessed using a 10-fold cross-validation procedure. A detailed description of the experimental setting, along with a brief discussion of the chosen hyperparameters, is provided in Appendix F.3.

The results are reported in Table 2, which shows the mean accuracy and standard deviation across the folds for each method at different coarsening ratios $r$. We observe that INGC consistently outperforms the SOTA methods on large datasets (Co-Physics, Pubmed, and Co-CS), and matches their performance on medium-sized datasets. SINGC also outperforms baseline methods in most settings, despite having a simpler optimization objective. A key takeaway from the results is that the integration of node features into the coarsening process gives FGC,INGC and SINGC a competitive advantage over methods that primarily focus on structural properties, such as SCAL and GCOND. This relationship is intuitive, as node classification relies not only on structural relationships but also on the meaningful preservation of node features.

## 5 CONCLUSION

In this paper, we introduced a novel graph coarsening method that focuses on preserving the inner products of graph signals during the coarsening process. We demonstrated that, although primarily considering node features, our approach also maintains the global structure of the graph. Our methods, INGC and SINGC, outperform SOTA techniques across various graph coarsening metrics and tasks like node classification, showcasing their versatility and effectiveness in preserving essential graph properties. These results highlight the potential of our approach for diverse graph-based learning applications. Future work includes implementing our coarsening method in graph pooling and evaluating its impact on improving GNN performance.

## 6 ETHICS STATEMENT

In this research, we exclusively used publicly available datasets for graph coarsening and node classification. Our work does not involve human subjects, personal data, or sensitive information.

We are committed to transparency, reproducibility, and the ethical use of machine learning techniques.

## 7 REPRODUCIBILITY STATEMENT

We ensure reproducibility by providing a detailed description of our methodology, including algorithmic steps (Algorithms 1, 2), evaluation procedures, and hyperparameter settings (Appendix F.2,F.3). The code used in this paper will be made available in a public repository upon acceptance. Full proofs of the theoretical results are included in the appendix, along with precise descriptions of our experimental setups.

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

# A  THEOREMS' PROOFS

## A.1  PROOF OF THEOREM 1

**Proof 1 (Proof of Theorem 1)** *Given a graph $\mathcal{G}$ with a graph Laplacian $\boldsymbol{L}$ and its coarsened graph $\mathcal{G}_c$ with a graph Laplacian $\boldsymbol{L}_c = \boldsymbol{C}^T \boldsymbol{L} \boldsymbol{C}$, and assume that for any two graph signals $\boldsymbol{x}, \boldsymbol{y} \in \mathbb{R}^n$ , the following condition is satisfied:*

$$\boldsymbol{x}^\top \boldsymbol{L} \boldsymbol{y} = \boldsymbol{x}_c^\top \boldsymbol{L}_c \boldsymbol{y}_c \tag{13}$$

*We plug in the the definitions of $\boldsymbol{x}_c = \boldsymbol{P}\boldsymbol{x}$ and $\boldsymbol{y}_c = \boldsymbol{P}\boldsymbol{y}$ in equation 13 and obtain:*

$$
\begin{aligned}
\boldsymbol{x}^\top \boldsymbol{L} \boldsymbol{y} &= \boldsymbol{x}_c^\top L_c \boldsymbol{y}_c \\
&= (P\boldsymbol{x})^\top \boldsymbol{L}_c \boldsymbol{P} \boldsymbol{y} \\
&= \boldsymbol{x}^\top \boldsymbol{P}^\top \boldsymbol{L}_c \boldsymbol{P} \boldsymbol{y} \\
&= \boldsymbol{x}^\top \boldsymbol{L}_l \boldsymbol{y}.
\end{aligned}
\tag{14}
$$

*where in the last equality we plug-in the definition of the lifted Laplacian (reconstructed Laplacian) $\boldsymbol{L}_l = \boldsymbol{P}^T \boldsymbol{L}_c \boldsymbol{P}$.*

*Assuming equation 14 holds for every pair of signals $\boldsymbol{x}, \boldsymbol{y} \in \mathbb{R}^n$, one can choose specific signals such that $\boldsymbol{L}[i,j] = \boldsymbol{L}_l[i,j]$ for all $i, j = 1, \ldots, n$, allowing us to conclude:*

$$\boldsymbol{L} = \boldsymbol{L}_l$$

*This implies that the full Laplacian $\boldsymbol{L}$ can be fully reconstructed for $\boldsymbol{L}_c$.*

## A.2  PROOF OF THEOREM 2

**Proof 2 (Proof of Theorem 2)** *Let $\boldsymbol{x}, \boldsymbol{y} \in \mathbb{R}^n$ be two k-smooth signals on the graph $\mathcal{G}$ with graph Laplacian $\boldsymbol{L}$. Define $\boldsymbol{x}_c = \boldsymbol{C}^\top \boldsymbol{x}, \boldsymbol{y}_c = \boldsymbol{C}^\top \boldsymbol{y} \in \mathbb{R}^k$, and let $\boldsymbol{L}_c = \boldsymbol{C}^\top \boldsymbol{L} \boldsymbol{C}$, where $\boldsymbol{C} = \boldsymbol{U}^{(k)} \boldsymbol{O}$, and $\boldsymbol{O} \in \mathcal{O}$ is some k-dimension rotation matrix. Then, the following relation holds:*

$$
\begin{aligned}
\boldsymbol{x}^\top \boldsymbol{L} \boldsymbol{y} - \boldsymbol{x}_c^\top \boldsymbol{L}_c \boldsymbol{y}_c &= \boldsymbol{x}^\top \boldsymbol{L} \boldsymbol{y} - ((\boldsymbol{C}^\top \boldsymbol{x})^\top \boldsymbol{C}^\top \boldsymbol{L} \boldsymbol{C} \boldsymbol{C}^\top \boldsymbol{y}) \\
&= \boldsymbol{x}^\top \boldsymbol{L} \boldsymbol{y} - ((\boldsymbol{U}^{(k)} \boldsymbol{O})^\top \boldsymbol{x})^\top (\boldsymbol{U}^{(k)} O)^\top \boldsymbol{L} \boldsymbol{U}^{(k)} \boldsymbol{O} (\boldsymbol{U}^{(k)} \boldsymbol{O})^\top \boldsymbol{y} \\
&= \boldsymbol{x}^\top \boldsymbol{L} \boldsymbol{y} - \boldsymbol{x}^\top \boldsymbol{U}^{(k)} \boldsymbol{O} \boldsymbol{O}^\top (\boldsymbol{U}^{(k)})^\top \boldsymbol{L} \boldsymbol{U}^{(k)} \boldsymbol{O} \boldsymbol{O}^\top (\boldsymbol{U}^{(k)})^\top \boldsymbol{y} \\
&= \boldsymbol{x}^\top \boldsymbol{L} \boldsymbol{y} - \boldsymbol{x}^\top \boldsymbol{U}^{(k)} (\boldsymbol{U}^{(k)})^\top L \boldsymbol{U}^{(k)} (\boldsymbol{U}^{(k)})^\top \boldsymbol{y} \\
&= \boldsymbol{x}^\top \boldsymbol{L} \boldsymbol{y} - \boldsymbol{x}^\top \boldsymbol{L} \boldsymbol{y} = 0
\end{aligned}
$$

*where the third equality holds because $\boldsymbol{O}$ is a rotation matrix satisfying $\boldsymbol{O}\boldsymbol{O}^\top = \boldsymbol{I}_{k \times k}$. The fifth equality holds because $\boldsymbol{x}$ and $\boldsymbol{y}$ are k-smooth and satisfy $\boldsymbol{x} = \boldsymbol{U}^{(k)}(\boldsymbol{U}^{(k)})^\top \boldsymbol{x}$ and $\boldsymbol{y} = \boldsymbol{U}^{(k)}(\boldsymbol{U}^{(k)})^\top \boldsymbol{y}$.*

*Thus, for any matrix $\boldsymbol{X}$ whose columns are k-smooth signals, we have:*

$$\|\boldsymbol{X}^\top \boldsymbol{L} \boldsymbol{X} - \boldsymbol{X}_c^\top \boldsymbol{L}_c \boldsymbol{X}_c\|_F^2 = 0.$$

## A.3  PROOF OF THEOREM 3 AND 4

The proof of Theorem 3 relies on the following definition and lemma.

**Definition 6 (Restricted spectral approximation Loukas & Vandergheynst (2018))** *Let $R$ be a k-dimensional subspace of $\mathbb{R}^n$. Matrices $\boldsymbol{L}_c$ and $\boldsymbol{L}$ are $(R, \epsilon)$-similar if there exists an $\epsilon > 0$ such that*

$$\|\boldsymbol{x} - \boldsymbol{x}_l\|_L \le \epsilon \|\boldsymbol{x}\|_L, \quad \text{for all } \boldsymbol{x} \in R,$$

*where $\boldsymbol{x}_l = \boldsymbol{C}\boldsymbol{C}^\dagger \boldsymbol{x}$.*

**Lemma 1** *Let $\boldsymbol{L}$ be the Laplacian matrix of a connected graph $\mathcal{G}$, and let $\boldsymbol{U}^{(k)}$ be the matrix containing its $k$-leading eigenvectors. Suppose $\boldsymbol{L}_c = \boldsymbol{C}^\top \boldsymbol{L} \boldsymbol{C}$ is the Laplacian matrix of a coarsened graph derived using a coarsening operator $\boldsymbol{C}$ with normalized columns such that*

$$tr(\boldsymbol{U}^{(k)}(\boldsymbol{U}^{(k)})^\top \boldsymbol{C}\boldsymbol{C}^\top) = k - \epsilon.$$

*Then, the matrices $L$ and $L_c$ are $(R, \epsilon\kappa)$-similar, where $\kappa = \frac{\lambda_{max}(\boldsymbol{L})}{\lambda_2(\boldsymbol{L})}$, and $R = span(\boldsymbol{U}^{(k)})$*

**Proof 3 (Proof of Lemma 1)** *We note the projection matrix defined by $\boldsymbol{C}$ as $\boldsymbol{\Pi}_C = \boldsymbol{C}\boldsymbol{C}^\top$. Given the trace condition $tr(\boldsymbol{U}^{(k)}(\boldsymbol{U}^{(k)})^\top \boldsymbol{\Pi}_C) = k - \epsilon$, we can express it as:*

$$tr(\boldsymbol{U}^{(k)}(\boldsymbol{U}^{(k)})^\top \boldsymbol{\Pi}_C) = tr((\boldsymbol{U}^{(k)})^\top \boldsymbol{\Pi}_C \boldsymbol{U}^{(k)}) = k - \epsilon.$$

*From this, we immediately obtain:*

$$tr((\boldsymbol{U}^{(k)})^\top (\boldsymbol{I} - \boldsymbol{\Pi}_C)\boldsymbol{U}^{(k)}) = \epsilon. \tag{15}$$

*This follows from the fact that:*

$$k = tr((\boldsymbol{U}^{(k)})^\top \boldsymbol{U}^{(k)}) = tr(\boldsymbol{U}^{(k)}(\boldsymbol{U}^{(k)})^\top) = tr(\boldsymbol{U}^{(k)} \boldsymbol{I}_{n \times n} \boldsymbol{U}^{(k)})$$
$$= tr((\boldsymbol{U}^{(k)})^\top \boldsymbol{\Pi}_C \boldsymbol{U}^{(k)}) + tr((\boldsymbol{U}^{(k)})^\top (\boldsymbol{I} - \boldsymbol{\Pi}_C)\boldsymbol{U}^{(k)}),$$

*where the first equality holds because $\boldsymbol{U}^{(k)}$ is orthonormal matrix.*

*Next, we express the term $\|\boldsymbol{x} - \boldsymbol{x}_l\|_L$. Since the columns of $\boldsymbol{C}$ are orthogonal, we can use $\boldsymbol{x}_l = \boldsymbol{C}\boldsymbol{P}\boldsymbol{x} = \boldsymbol{C}\boldsymbol{C}^\dagger \boldsymbol{x} = \boldsymbol{C}\boldsymbol{C}^\top \boldsymbol{x}$, and obtain:*

$$\|\boldsymbol{x} - \boldsymbol{x}_l\|_L = (\boldsymbol{x} - \boldsymbol{C}\boldsymbol{C}^\top \boldsymbol{x})^\top \boldsymbol{L}(\boldsymbol{x} - \boldsymbol{C}\boldsymbol{C}^\top \boldsymbol{x})$$
$$= ((\boldsymbol{I} - \boldsymbol{C}\boldsymbol{C}^\top)\boldsymbol{x})^\top \boldsymbol{L}(\boldsymbol{I} - \boldsymbol{C}\boldsymbol{C}^\top)\boldsymbol{x}. \tag{16}$$

*Using the Rayleigh quotient (Spielman, 2019), we can bound by:*

$$\|\boldsymbol{x} - \boldsymbol{x}_l\|_L = ((\boldsymbol{I} - \boldsymbol{C}\boldsymbol{C}^\top)\boldsymbol{x})^\top \boldsymbol{L}((\boldsymbol{I} - \boldsymbol{C}\boldsymbol{C}^\top)\boldsymbol{x})$$
$$\leq \lambda_{max}(\boldsymbol{L})\|(\boldsymbol{I} - \boldsymbol{C}\boldsymbol{C}^\top)\boldsymbol{x}\|_2^2. \tag{17}$$

*Next, we proceed to bound the term $\|(\boldsymbol{I} - \boldsymbol{C}\boldsymbol{C}^\top)\boldsymbol{x}\|_2^2$. Since $\boldsymbol{x}$ is spanned by $\boldsymbol{U}^{(k)}$, we write $\boldsymbol{x} = \boldsymbol{U}^{(k)}\boldsymbol{z}$. Therefore, we get:*

$$\|(\boldsymbol{I} - \boldsymbol{C}\boldsymbol{C}^\top)\boldsymbol{x}\|_2^2 = \boldsymbol{z}^\top (\boldsymbol{U}^{(k)})^\top (\boldsymbol{I} - \boldsymbol{C}\boldsymbol{C}^\top)\boldsymbol{U}^{(k)}\boldsymbol{z}.$$

*From equation 15, we know that the maximum eigenvalue of $(\boldsymbol{U}^{(k)})^\top (\boldsymbol{I} - \boldsymbol{\Pi}_C)\boldsymbol{U}^{(k)}$ is bounded by $\epsilon$. Thus, by applying the Rayleigh quotient, we obtain:*

$$\|(\boldsymbol{I} - \boldsymbol{C}\boldsymbol{C}^\top)\boldsymbol{x}\|_2^2 \leq \epsilon\|\boldsymbol{z}\|_2^2 = \epsilon\|\boldsymbol{x}\|_2^2.$$

*Substituting this bound into equation 17, we have:*

$$\|\boldsymbol{x} - \boldsymbol{x}_l\|_L \leq \epsilon\lambda_{max}(\boldsymbol{L})\|\boldsymbol{x}\|_2^2.$$

*Since $\boldsymbol{L}$ is the graph Laplacian of a connected graph, it has only one zero eigenvalue, corresponding to the constant vector. Assuming $\boldsymbol{x}$ is not a constant vector, we can bound $\|\boldsymbol{x}\|_2^2$ using the Rayleigh quotient:*

$$\|\boldsymbol{x}\|_2^2 \geq \frac{\|\boldsymbol{x}\|_L}{\lambda_2(\boldsymbol{L})}.$$

*Substituting this into the previous inequality, we obtain:*

$$\|\boldsymbol{x} - \boldsymbol{x}_l\|_L \leq \epsilon \frac{\lambda_{max}(\boldsymbol{L})}{\lambda_2(\boldsymbol{L})}\|\boldsymbol{x}\|_L = \epsilon\kappa\|\boldsymbol{x}\|_L,$$

*where $\kappa = \frac{\lambda_{max}(\boldsymbol{L})}{\lambda_2(\boldsymbol{L})}$ is the condition number of L.*

*Finally, if $\boldsymbol{x}$ is a constant vector, then since the columns of $\boldsymbol{C}$ span the constant vector, we have:*

$$\|\boldsymbol{x} - \boldsymbol{x}_l\|_L = \|\boldsymbol{x} - \boldsymbol{C}\boldsymbol{C}^\top\boldsymbol{x}\|_L = 0.$$

*Thus, for all $\boldsymbol{x} \in span(U^{(k)})$, we conclude that:*

$$\|\boldsymbol{x} - \boldsymbol{x}_l\|_L \leq \epsilon\kappa\|\boldsymbol{x}\|_L.$$

*i.e $\boldsymbol{L}$ and $\boldsymbol{L}_c$ are $(R, \epsilon\kappa)$ similar.*

Then, according to Theorem 13 and Corollary 12 in (Loukas, 2019), if the full graph Laplacian $\boldsymbol{L}$ and the coarsen graph Laplacian are $\boldsymbol{L}_c$ are $(\boldsymbol{U}^{(k)}, \epsilon\kappa)$-similar, they satisfy the following inequalities: :

$$(1 - \epsilon\kappa)\|\boldsymbol{x}\|_L \leq \|\boldsymbol{x}\|_{L_c} \leq (1 + \epsilon\kappa)\|\boldsymbol{x}\|_L$$

$$\frac{1}{\mu_1}\lambda^{(i)} \leq \lambda_c^{(i)} \leq \frac{1}{\mu_2}\frac{(1 + \epsilon\kappa)^2}{1 - (\epsilon\kappa)^2(\lambda^{(i)}/\lambda^{(2)})}\lambda^{(i)}, \qquad 2 \leq i \leq k$$

according to Theorem 3, where $\kappa = \frac{\lambda_{max}(\boldsymbol{L})}{\lambda_2(\boldsymbol{L})}$, and $\mu_1, \mu_2$ and the first and $k$ eigenvalues of the matrix $\boldsymbol{P}\boldsymbol{P}^\top$.

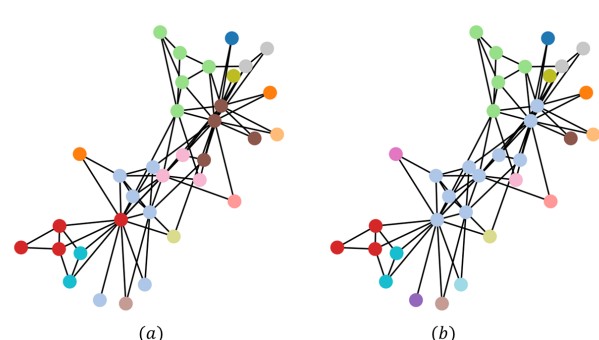

(a)    (b)

Figure 2: Clustering of the Karate Club network using (a) our SINGC method and (b) conventional $k$-means clustering on the leading eigenvectors. Each color represents a distinct cluster, and the coarsened graph is obtained by mapping nodes from the same cluster (color) to a single super-node. We observe that the clusters produced by SINGC are more balanced, which is advantageous for downstream graph learning tasks.

### A.4 GRADIENT COMPUTATION

We start with a recap of our suggested objective function:

$$\min_{\boldsymbol{X}_c, \boldsymbol{C}} f(\boldsymbol{X}_c, \boldsymbol{C}) = \|\boldsymbol{X}^\top\boldsymbol{L}\boldsymbol{X} - \boldsymbol{X}_c^\top\boldsymbol{C}^\top\boldsymbol{L}\boldsymbol{C}\boldsymbol{X}_c\|_F - \beta tr(\boldsymbol{U}^{(k)}(\boldsymbol{U}^{(k)})^\top\boldsymbol{C}\boldsymbol{C}^\top)$$

$$+ \lambda\|\boldsymbol{C}^T\|_{1,2}^2 - \alpha logdet(\boldsymbol{L}_c + \boldsymbol{J})$$

$$\textbf{s.t.} \qquad \boldsymbol{X}_c = \boldsymbol{C}^\dagger\boldsymbol{X}$$

The gradient of each term with respect to $C$ is:

$$\nabla_{\boldsymbol{C}}(-tr(\boldsymbol{U}^{(k)}(\boldsymbol{U}^{(k)})^\top \boldsymbol{C}\boldsymbol{C}^\top)) = -2\boldsymbol{U}^{(k)}(\boldsymbol{U}^{(k)})^\top \boldsymbol{C}$$

$$\nabla_{\boldsymbol{C}}(\|\boldsymbol{X}^\top \boldsymbol{LX} - \boldsymbol{X}_c^\top \boldsymbol{C}^\top \boldsymbol{LCX}_c\|_F) = -(2\boldsymbol{LCX}_c(\boldsymbol{X}^\top \boldsymbol{LX} - (\boldsymbol{LCX}_c)^\top \boldsymbol{CX}_c)\boldsymbol{X}_c^\top$$
$$+ 2\boldsymbol{L}^\top \boldsymbol{CX}_c(\boldsymbol{X}^\top \boldsymbol{LX} - (\boldsymbol{CX}_c)^\top \boldsymbol{LCX}_c)\boldsymbol{X}_c^\top)$$

$$\nabla_{\boldsymbol{C}}(\|\boldsymbol{C}^T\|_{1,2}^2) = \boldsymbol{C}\mathbf{1}_{k\times k}$$

$$\nabla_{\boldsymbol{C}}(logdet(\boldsymbol{L}_c + \boldsymbol{J})) = \boldsymbol{LC}(\boldsymbol{C}^\top \boldsymbol{LC} + \boldsymbol{J})^{-1}$$

where the third was shown in (Kumar et al., 2023), assuming all elements of $\boldsymbol{C}$ to non-negative (since $\boldsymbol{C} \in \mathcal{C}$. The full gradient with respect to $\boldsymbol{C}$ of equation 11 is:

$$\nabla_{\boldsymbol{C}} f(\boldsymbol{C}, \boldsymbol{X}_c) = 2\beta \boldsymbol{U}^{(k)}(\boldsymbol{U}^{(k)})^\top \boldsymbol{C} + \lambda \boldsymbol{C}$$
$$- (2\boldsymbol{LCX}_c(\boldsymbol{X}^\top \boldsymbol{LX} - (\boldsymbol{LCX}_c)^\top \boldsymbol{CX}_c)\boldsymbol{X}_c^\top$$
$$+ 2\boldsymbol{L}^\top \boldsymbol{CX}_c(\boldsymbol{X}^\top \boldsymbol{LX} - (\boldsymbol{CX}_c)^\top \boldsymbol{LCX}_c)\boldsymbol{X}_c^\top)$$
$$+ \lambda \boldsymbol{C}\mathbf{1}_{k\times k} - \alpha(\boldsymbol{LC}(\boldsymbol{C}^\top \boldsymbol{LC} + \boldsymbol{J})^{-1}) \tag{18}$$

We note that in case of the SINGC Algorithm the computed gradient is simpler and can be express as:

$$\nabla_{\boldsymbol{C}} f(\boldsymbol{C}, \boldsymbol{X}_c) = 2\boldsymbol{U}^{(k)}(\boldsymbol{U}^{(k)})^\top \boldsymbol{C} + \lambda \boldsymbol{C}\mathbf{1}_{k\times k} - \alpha(\boldsymbol{LC}(\boldsymbol{C}^\top \boldsymbol{LC} + \boldsymbol{J})^{-1}) \tag{19}$$

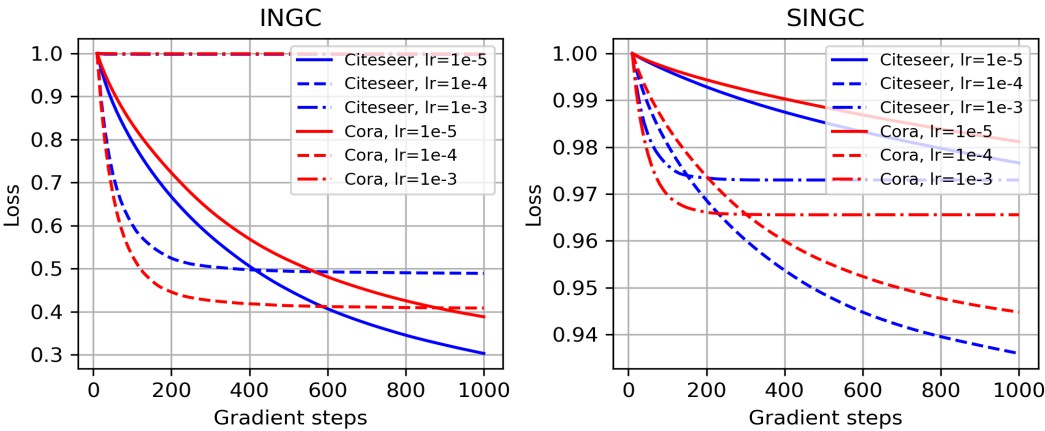

Figure 3: Convergence rates of INGC and SINGC methods for Cora and Citeseer datasets. Citeseer results are shown in blue with varying line styles for different learning rates, while Cora results are shown in red. Gradient steps are on the x-axis, and normalized loss values are on the y-axis.

## B  METHODS PERFORMANCE - VISUAL COMPARISON

In Section 4.1, we demonstrated the global preservation property of our method on a specific task with a low number of super-nodes, similar to a clustering task. However, in typical coarsening scenarios, there are usually a larger number of super-nodes. In this section, we present the performance of this property in more practical coarsening scenarios, showing how our method continues to preserve the global structure of the graph.

In Figures 5 and 6 , we present the results obtained by our methods (INGC/SINGC), alongside the three baseline methods described in Section 4.1 on the Karate Club and Les Miserables datasets. Each row in the figures shows the partitioning produced by each method at different coarsening ratios. Nodes of the same color are grouped into the same super-node. We observe that our method groups adjacent nodes into super-nodes, thereby preserving the global structure of the graph.

Since our method leverages the graph Laplacian eigenvectors to partition the vertices, we also compare it to the commonly used spectral clustering approach (Von Luxburg, 2007), which partitions

the vertices by applying k-means (Jain & Dubes, 1988) on the leading graph Laplacian eigenvectors. In Figure 2, we present the clustering results obtained by the SINGC method on the well-known Karate Club dataset(Zachary, 1977), which consists of $n = 34$ nodes. We apply our method with a target of $k = \frac{n}{2} = 17$ super-nodes and compare the results to those obtained by spectral clustering(Von Luxburg, 2007) applied to the top $k$ leading eigenvectors. In the figure, each color represents a distinct cluster.We observe that the clusters produced by our method are more balanced compared to those generated by spectral clustering, which tends to form one large cluster alongside several smaller, single-node clusters. This balance is advantageous for downstream graph learning tasks, such as graph pooling.

Ron et al. (2011)

## C  COMPLEXITY ANALYSIS

For an input graph with $n$ nodes, $e$ edges, and a feature vector of size $p$ for each node, the coarsened graph has $k$ nodes and $e_c$ edges. The gradient computation per iteration primarily drives the computational cost of coarsening optimization. Table 3 summarizes the gradient expressions and their time complexities, highlighting that SINGC is the most efficient, while INGC remains competitive with FGC.

Regarding incorporating coarsening in GCNs, the total time complexity for node classification on the original graph is $O(n^2 lp + nle)$. Since the number of coarsened nodes $k$ is typically greater than the number of node features $p$, applying coarsening before a GCN is particularly beneficial for dense graphs where $e > n$. Coarsening reduces the graph size while maintaining the dominant complexity term at $O(n^2)$, ensuring efficiency for large-scale graphs.

| | FGC | INGC | SINGC |
|---|---|---|---|
| **Gradient Expression** | $\nabla_C f(C, X_c) = 2((CX_c - X) + L(CX_c))X_c^\top + \lambda C \mathbf{1}_{k \times k} - \alpha(LC(C^\top LC + J)^{-1})$ | $\nabla_C f(C, X_c) = 2\beta U^{(k)}(U^{(k)})^\top C - [2L(CX_c)(X^\top LX - (LCX_c)^\top (CX_c))X_c^\top] + \lambda C \mathbf{1}_{k \times k} - \alpha(LC(C^\top LC + J)^{-1})$ | $\nabla_C f(C) = 2U^{(k)}(U^{(k)})^\top C + \lambda C \mathbf{1}_{k \times k} - \alpha(LC(C^\top LC + J)^{-1})$ |
| **Theoretical Time Complexity** | $O(n^2(k+d) + k^3)$ | $O(n^2(k+d) + ndk + nk^2 + k^3)$ | $O(n^2 k + nk^2 + k^3)$ |

Table 3: Comparison of gradient expressions and time complexities for FGC, INGC, and SINGC.

## D  CONVERGENCE ANALYSIS

Figure 3 illustrates the convergence rates of the INGC and SINGC methods on the Cora dataset for a coarsening ratio $r = 0.3$. The left subplot shows the performance of INGC, while the right subplot depicts SINGC. For both methods, Citeseer results are in blue with varying line styles for different learning rates, and Cora results in red with corresponding line styles. The x-axis represents gradient steps, and the y-axis shows normalized loss values.

The results reveal a typical convergence pattern for different learning. A trade-off is observed between convergence speed and final objective loss: higher learning rates lead to faster convergence but result in a higher final loss. This phenomenon is consistent across both datasets. We note that recent work has shown that lower learning rates can achieve lower minimal loss values but may risk unstable solutions Mulayoff et al. (2021).

## E  HYPER PARAMETERS DISCUSSION AND ABLATION STUDY

We review the purpose of each term in our optimization and clarify the motivations behind selecting the hyperparameters values. The parameter $\beta$ promotes minimizing the IPE for general smooth signals. As shown in Theorem 3, minimizing the respective term also bounds the REE (related to

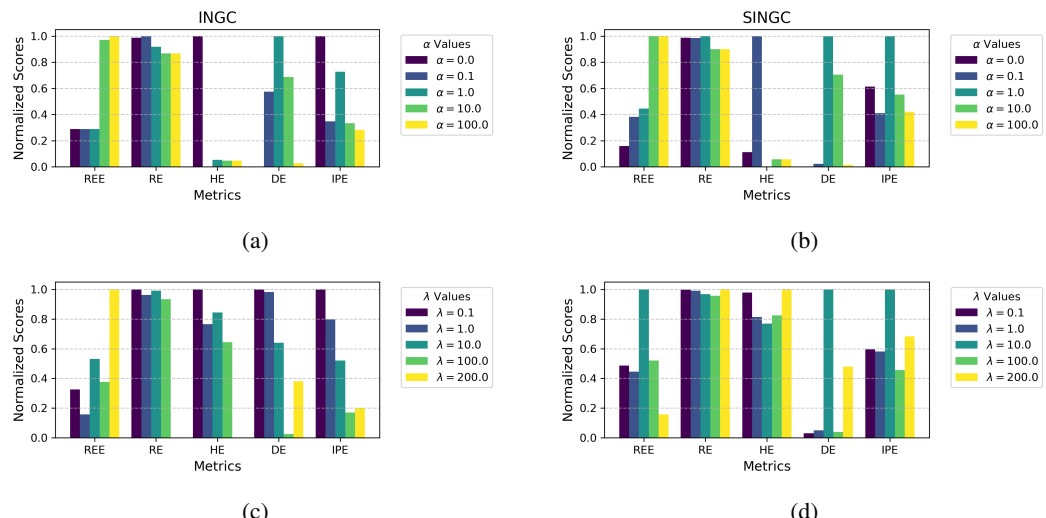

(a)                      (b)

(c)                      (d)

Figure 4: Ablation study on the sensitivity and contribution of the hyperparameters $\alpha$ and $\lambda$ across different metrics on the Cora dataset with a coarsening ratio $r = 0.3$. (a) and (b) show the sensitivity of the parameter $\alpha$ across metrics. (c) and (d) illustrate the sensitivity of the methods to parameter $\lambda$. The bars represent normalized scores for different values of the respective hyperparameter, with distinct colors denoting specific values. Lower bar values indicate better performance.

| Dataset | r | INGC($\beta = 0$) | INGC | SINGC |
|---------|-----|-------------------|------|-------|
| | 0.3 | $84.62 \pm 0.59$ | $\mathbf{87.55 \pm 0.16}$ | $84.51 \pm 0.33$ |
| Cora | 0.1 | $83.01 \pm 0.53$ | $\mathbf{83.38 \pm 0.47}$ | $\underline{82.76 \pm 0.32}$ |
| | 0.05 | $76.92 \pm 1.11$ | $\underline{77.42 \pm 0.78}$ | $\mathbf{77.81 \pm 0.68}$ |
| | 0.3 | $76.25 \pm 0.28$ | $\mathbf{76.89 \pm 0.23}$ | $\underline{76.66 \pm 0.27}$ |
| Citeseer | 0.1 | $67.07 \pm 0.59$ | $\mathbf{72.63 \pm 0.25}$ | $\underline{69.71 \pm 0.72}$ |
| | 0.05 | $60.66 \pm 1.58$ | $\underline{66.02 \pm 0.32}$ | $\mathbf{66.37 \pm 0.57}$ |
| | 0.05 | $\mathbf{83.60 \pm 0.23}$ | $\underline{83.59 \pm 0.22}$ | $83.55 \pm 0.32$ |
| Pubmed | 0.03 | $81.62 \pm 0.14$ | $\underline{81.93 \pm 0.22}$ | $\mathbf{83.19 \pm 0.18}$ |
| | 0.01 | $79.08 \pm 0.72$ | $\underline{79.09 \pm 0.26}$ | $\mathbf{79.96 \pm 0.34}$ |
| | 0.05 | $90.42 \pm 0.18$ | $\underline{90.84 \pm 0.12}$ | $\mathbf{90.92 \pm 0.22}$ |
| Co-CS | 0.03 | $89.28 \pm 0.21$ | $\underline{89.59 \pm 0.38}$ | $\mathbf{89.99 \pm 0.41}$ |
| | 0.01 | $77.79 \pm 1.15$ | $\mathbf{87.93 \pm 0.33}$ | $\underline{83.39 \pm 0.33}$ |
| #Best | | 1 | 6 | 6 |
| #2-Best | | 1 | 6 | 5 |

Table 4: Ablation study of the parameter $\beta$ on node classification tasks. The table reports the accuracy on various datasets for different coarsening ratios $r$ using different coarsening methods. The third column presents the results of our INGC method with $\beta = 0$, the fourth column corresponds to the optimal $\beta$ value, and the fifth column shows the results of SINGC. Best results are in bold; second-best results are underlined. The last two rows indicate the number of times each method achieved the best and second-best performance.

preserving the graph's global structure) and DE (related to preserving the norm of node features). Therefore, $\beta$ is significant when these properties are prioritized in coarsening. The parameter $\lambda$ enforces group sparsity in each row, ensuring the validity of the obtained coarsening operator $C$. Since $C$ lacks meaningful structure without this term, we did not perform an ablation study on $\lambda$. Finally, the parameter $\alpha$ promotes connectivity in the coarsened graph, making it significant in scenarios where preserving graph connectivity is essential.

Figure 4 presents an experiment analyzing each parameter's contribution and our method's sensitivity to their variations. The bars represent normalized scores for different values of a given

hyperparameter, with distinct colors denoting specific values. For all metrics, lower values indicate better performance. The other two parameters are set to their optimal values for each metric as specified in Table 7. The figure illustrates the sensitivity of each parameter and evaluates the impact of deviations from optimal values on various metrics.

In Figures 4(a) and 4(b), varying $\alpha$ shows minimal sensitivity across metrics, except for IPE, where changes up to an order of magnitude still yield similar results. Additionally, the figures include an ablation study on the parameter $\alpha$ illustrating its contribution to the optimization process. In Figures 4(c) and 4(d), varying $\lambda$ demonstrates that our methods are more sensitive to this parameter, highlighting its critical role in performance.

In Table 2, we present an ablation study on the parameter $\beta$ for the node classification task across various datasets and coarsening ratios $r$. The comparison includes three methods: INGC with $\beta = 0$ (ignoring the term $\text{tr}(U^{(k)}(U^{(k)})^\top CC^\top)$ for minimizing IPE for general smooth signals), INGC with the optimal $\beta$, and SINGC (our second proposed method, which omits the first term of the objective entirely). The table reports node classification accuracy, with the best results highlighted in bold and the second-best results underlined. For each metric, other hyperparameters are set to their optimal values. The results demonstrate the importance of balancing the two complementary approaches to minimizing IPE. INGC with $\beta = 0$ generally underperforms compared to the other methods, emphasizing the significance of the smooth signal term in achieving high classification accuracy.

# F  ADDITIONAL DETAILS ON THE EXPERIMENTAL STUDY

## F.1  DATASETS DETAILS

The additional details of real datasets are as follows:

- **Karate Club** - $n = 34$, $p = 30$, $|\mathcal{E}| = 78$ - Here, nodes represent members of a karate club, and edges represent friendships between them. Synthetic features generated using the signal model presented at Section 3.3.

- **Les Miserables** - $n = 77$, $p = 50$, $|\mathcal{E}| = 254$ - Nodes represent characters in the novel *Les Miserables*, and edges indicate co-occurrence in the same chapter. Synthetic features generated using the signal model presented at Section 3.3.

- **Cora** - $n = 2,708$, $p = 1,433$, $|\mathcal{E}| = 5,429$ - Nodes represent research papers, and edges represent citation links between them. Node features correspond to the presence of specific words in each paper, and class labels indicate the paper's research field. Number of classes = 7.

- **Citeseer** - $n = 3,327$, $p = 3,703$, $|\mathcal{E}| = 4,732$ - Nodes represent research papers, and edges represent citation relationships. Node features are based on word occurrences in each paper, and class labels indicate the paper's topic. Number of classes = 6.

- **Co-Physics** - $n = 34,493$, $p = 8,415$, $|\mathcal{E}| = 247,962$ - Nodes represent physics research papers, and edges represent citations. Node features represent article keywords, and class labels indicate different fields of physics. Number of classes = 5.

- **PubMed** - $n = 19,717$, $p = 500$, $|\mathcal{E}| = 44,338$ - Nodes represent biomedical research papers, and edges represent citations. Node features are derived from TF-IDF scores of medical terms, and class labels indicate disease categories. Number of classes = 3.

- **Co-Computer** - $n = 13,752$, $p = 767$, $|\mathcal{E}| = 245,861$ - Nodes represent products in a co-purchase network, and edges indicate products frequently purchased together. Node features describe product attributes, and class labels represent product categories. Number of classes = 10.

- **Co-CS** - $n = 18,333$, $p = 7005$ , $|\mathcal{E}| = 163,788$ - Here, nodes are authors, that are connected by an edge if they co-authored a paper; node features represent paper keywords for each author's papers, and class labels indicate most active fields of study for each author. Number of classes = 15.

## F.2 GRAPH COARSENING METRICS EXPERIMENTS SETTING

Tables 5, 6, 7, and 8 present the hyperparameters of our methods for each experiment. For SINGC, we set $t_{\text{iter}} = 2000$ in all experiments, and for INGC, we set $t_{\text{iter}} = 20$ and $c_{\text{iter}} = 100$.

Regarding the implementation of the baseline comparison methods, the FGC hyperparameters were selected based on their optimal values as reported in their paper. The LVN and LVE methods were implemented using their provided graph coarsening libraries, with the maximum value of the parameter $K = k = r \cdot n$.

| Metric | Method | Karate Club dataset | | |
|--------|--------|---------------------|---|---|
| | | $r = 0.7$ | $r = 0.5$ | $r = 0.3$ |
| REE | INGC | $\beta$=0, $\lambda$=100, $\alpha$=0.1 | $\beta$=200, $\lambda$=10, $\alpha$=1 | $\beta$=100, $\lambda$=100, $\alpha$=0.1 |
| | SINGC | $\lambda$=0.1, $\alpha$=0.1 | $\lambda$=0.1, $\alpha$=0.1 | $\lambda$=0.1, $\alpha$=0.1 |
| RE | INGC | $\beta$=0, $\lambda$=1, $\alpha$=200 | $\beta$=200, $\lambda$=0.1, $\alpha$=200 | $\beta$=200, $\lambda$=1, $\alpha$=200 |
| | SINGC | $\lambda$=0.1, $\alpha$=200 | $\lambda$=0.1, $\alpha$=200 | $\lambda$=1, $\alpha$=200 |
| HE | INGC | $\beta$=0, $\lambda$=1, $\alpha$=200 | $\beta$=200, $\lambda$=0.1, $\alpha$=200 | $\beta$=200, $\lambda$=1, $\alpha$=200 |
| | SINGC | $\lambda$=0.1, $\alpha$=200 | $\lambda$=0.1, $\alpha$=200 | $\lambda$=10, $\alpha$=200 |
| DEE | INGC | $\beta$=0, $\lambda$=1, $\alpha$=200 | $\beta$=200, $\lambda$=1, $\alpha$=200 | $\beta$=0.1, $\lambda$=1, $\alpha$=200 |
| | SINGC | $\lambda$=10, $\alpha$=200 | $\lambda$=1, $\alpha$=200 | $\lambda$=200, $\alpha$=200 |

Table 5: Graph coarsening metrics experimental setting: Chosen hyperparameters for the Karate Club dataset at different coarsening ratios ($r$) and metrics.

| Metric | Method | Les Miserables dataset | | |
|--------|--------|------------------------|---|---|
| | | $r = 0.7$ | $r = 0.5$ | $r = 0.3$ |
| REE | INGC | $\beta$=100, $\lambda$=200, $\alpha$=0.1 | $\beta$=200, $\lambda$=10, $\alpha$=0.1 | $\beta$=200, $\lambda$=200, $\alpha$=1 |
| | SINGC | $\lambda$=0.1, $\alpha$=0.1 | $\lambda$=100, $\alpha$=0.1 | $\lambda$=0.1, $\alpha$=0.1 |
| RE | INGC | $\beta$=200, $\lambda$=10, $\alpha$=100 | $\beta$=200, $\lambda$=10, $\alpha$=200 | $\beta$=200, $\lambda$=100, $\alpha$=200 |
| | SINGC | $\lambda$=0.1, $\alpha$=100 | $\lambda$=1, $\alpha$=200 | $\lambda$=100, $\alpha$=100 |
| HE | INGC | $\beta$=200, $\lambda$=10, $\alpha$=100 | $\beta$=200, $\lambda$=10, $\alpha$=200 | $\beta$=200, $\lambda$=100, $\alpha$=200 |
| | SINGC | $\lambda$=0.1, $\alpha$=100 | $\lambda$=0.1, $\alpha$=200 | $\lambda$=100, $\alpha$=100 |
| DEE | INGC | $\beta$=0, $\lambda$=200, $\alpha$=200 | $\beta$=0.1, $\lambda$=0.1, $\alpha$=10 | $\beta$=0.1, $\lambda$=0.1, $\alpha$=200 |
| | SINGC | $\lambda$=100, $\alpha$=100 | $\lambda$=10, $\alpha$=100 | $\lambda$=100, $\alpha$=200 |

Table 6: Graph coarsening metrics experimental setting: Chosen hyperparameters for the Les Miserables dataset at different coarsening ratios ($r$) and metrics.

## F.3 NODE CLASSIFICATION EXPERIMENTS SETTING

The GCN model used in our experiments consists of two graph convolutional layers and is implemented using PyTorch and PyTorch Geometric libraries. The architecture is as follows:

- **Layer 1:** A Graph Convolutional Network (GCNConv) layer that takes the input node feature matrix $X$ (with $X.shape[1]$ features) and outputs a hidden representation of size 64.
- **Layer 2:** A second GCNConv layer that maps the 64-dimensional hidden representation to the number of output classes (NUM_OF_CLASSES).

We use ReLU for non-linearity and dropout for regularization during training.

| Metric | Method | Cora dataset | | |
|---|---|---|---|---|
| | | $r = 0.7$ | $r = 0.5$ | $r = 0.3$ |
| REE | INGC | $\beta{=}10, \lambda{=}1, \alpha{=}1$ | $\beta{=}100, \lambda{=}1, \alpha{=}1$ | $\beta{=}200, \lambda{=}10, \alpha{=}10$ |
| | SINGC | $\lambda{=}1, \alpha{=}10$ | $\lambda{=}100, \alpha{=}100$ | $\lambda{=}200, \alpha{=}0.1$ |
| RE | INGC | $\beta{=}10, \lambda{=}100, \alpha{=}200$ | $\beta{=}100, \lambda{=}200, \alpha{=}200$ | $\beta{=}100, \lambda{=}10, \alpha{=}200$ |
| | SINGC | $\lambda{=}100, \alpha{=}200$ | $\lambda{=}0.1, \alpha{=}100$ | $\lambda{=}100, \alpha{=}100$ |
| HE | INGC | $\beta{=}10, \lambda{=}100, \alpha{=}200$ | $\beta{=}100, \lambda{=}200, \alpha{=}200$ | $\beta{=}100, \lambda{=}10, \alpha{=}200$ |
| | SINGC | $\lambda{=}100, \alpha{=}200$ | $\lambda{=}1, \alpha{=}10$ | $\lambda{=}100, \alpha{=}100$ |
| DEE | INGC | $\beta{=}0.1, \lambda{=}0.1, \alpha{=}10$ | $\beta{=}0.1, \lambda{=}100, \alpha{=}200$ | $\beta{=}0, \lambda{=}10, \alpha{=}10$ |
| | SINGC | $\lambda{=}100, \alpha{=}200$ | $\lambda{=}10, \alpha{=}100$ | $\lambda{=}100, \alpha{=}200$ |

Table 7: Graph coarsening metrics experimental setting: Chosen hyperparameters for the Cora dataset at different coarsening ratios ($r$) and metrics.

| Metric | Method | Citeseer dataset | | |
|---|---|---|---|---|
| | | $r = 0.7$ | $r = 0.5$ | $r = 0.3$ |
| REE | INGC | $\beta{=}200, \lambda{=}200, \alpha{=}200$ | $\beta{=}100, \lambda{=}10, \alpha{=}0.1$ | $\beta{=}100, \lambda{=}10, \alpha{=}0.1$ |
| | SINGC | $\lambda{=}10, \alpha{=}0.1$ | $\lambda{=}100, \alpha{=}0.1$ | $\lambda{=}10, \alpha{=}1$ |
| RE | INGC | $\beta{=}100, \lambda{=}10, \alpha{=}200$ | $\beta{=}100, \lambda{=}1, \alpha{=}10$ | $\beta{=}0, \lambda{=}10, \alpha{=}0.1$ |
| | SINGC | $\lambda{=}1, \alpha{=}100$ | $\lambda{=}10, \alpha{=}100$ | $\lambda{=}100, \alpha{=}200$ |
| HE | INGC | $\beta{=}100, \lambda{=}10, \alpha{=}200$ | $\beta{=}0.1, \lambda{=}100, \alpha{=}200$ | $\beta{=}200, \lambda{=}0.1, \alpha{=}0.1$ |
| | SINGC | $\lambda{=}1, \alpha{=}100$ | $\lambda{=}1, \alpha{=}100$ | $\lambda{=}100, \alpha{=}200$ |
| DEE | INGC | $\beta{=}1, \lambda{=}0.1, \alpha{=}0.1$ | $\beta{=}200, \lambda{=}1, \alpha{=}10$ | $\beta{=}0.1, \lambda{=}0.1, \alpha{=}10$ |
| | SINGC | $\lambda{=}100, \alpha{=}200$ | $\lambda{=}100, \alpha{=}200$ | $\lambda{=}100, \alpha{=}100$ |

Table 8: Graph coarsening metrics experimental setting: Chosen hyperparameters for the Citeseer dataset at different coarsening ratios ($r$) and metrics.

For SINGC, we set $t_{\text{iter}} = 2000$ in all experiments, and for INGC, we set $t_{\text{iter}} = 20$ and $c_{\text{iter}} = 100$. Tables 10 and 9 present the hyperparameters of our methods for each experiment. The results for the three baseline methods presented in Table 2 are sourced from Kumar et al. (2023).

We note that tuning the hyperparameters in our methods is crucial for achieving optimal performance. By reviewing some of the corresponding setting in Tables 8, 7 and 9 we can observe that good performance often aligns with low values of REE and INP in this application. Therefore, we recommend that practitioners first optimize the hyperparameters by minimizing REE and INP. Once optimized, the coarsened graph can be used in the GNN for training and evaluation, leading to improved classification accuracy.

| Dataset | Method | Node Classification Parameters - Medium datasets | | |
|---|---|---|---|---|
| | | $r = 0.3$ | $r = 0.1$ | $r = 0.05$ |
| Cora | INGC | $\beta{=}100, \lambda{=}100, \alpha{=}10$ | $\beta{=}1, \lambda{=}100, \alpha{=}1$ | $\beta{=}1, \lambda{=}100, \alpha{=}100$ |
| | SINGC | $\lambda{=}10, \alpha{=}0.01$ | $\lambda{=}1000, \alpha{=}1$ | $\lambda{=}1000, \alpha{=}0.01$ |
| Citeseer | INGC | $\beta{=}0, \lambda{=}100, \alpha{=}10$ | $\beta{=}10, \lambda{=}1000, \alpha{=}1000$ | $\beta{=}0, \lambda{=}20, \alpha{=}10$ |
| | SINGC | $\lambda{=}50, \alpha{=}20$ | $\lambda{=}300, \alpha{=}100$ | $\lambda{=}50, \alpha{=}10$ |

Table 9: Node classification experimental setting: Chosen hyperparameters for the Cora Citeseer dataset at different coarsening ratios ($r$) and metrics.

| Dataset | Method | Node Classification Parameters - Large datasets | | |
|---------|--------|---------------------------------------|---------------------------------------|---------------------------------------|
| | | $r = 0.05$ | $r = 0.03$ | $r = 0.01$ |
| Co-phy | INGC | $\beta{=}10, \lambda{=}10, \alpha{=}0.01$ | $\beta{=}10, \lambda{=}1000, \alpha{=}0.01$ | $\beta{=}10, \lambda{=}1000, \alpha{=}100$ |
| | SINGC | $\lambda{=}100, \alpha{=}0.01$ | $\lambda{=}10, \alpha{=}1$ | $\lambda{=}10, \alpha{=}100$ |
| Pubmed | INGC | $\beta{=}0, \lambda{=}1000, \alpha{=}0.001$ | $\beta{=}0.1, \lambda{=}100, \alpha{=}10$ | $\beta{=}0.1, \lambda{=}100, \alpha{=}100$ |
| | SINGC | $\lambda{=}1000, \alpha{=}0.001$ | $\lambda{=}100, \alpha{=}0.1$ | $\lambda{=}10, \alpha{=}10$ |
| Co-CS | INGC | $\beta{=}1, \lambda{=}1000, \alpha{=}10$ | $\beta{=}0.1, \lambda{=}1, \alpha{=}10$ | $\beta{=}10, \lambda{=}100, \alpha{=}100$ |
| | SINGC | $\lambda{=}40, \alpha{=}10$ | $\lambda{=}10, \alpha{=}10$ | $\lambda{=}100, \alpha{=}100$ |

Table 10: Node classification experimental setting: Chosen hyperparameters for the Co-phy, Pubmed, Co-CS dataset at different coarsening ratios ($r$) and metrics.

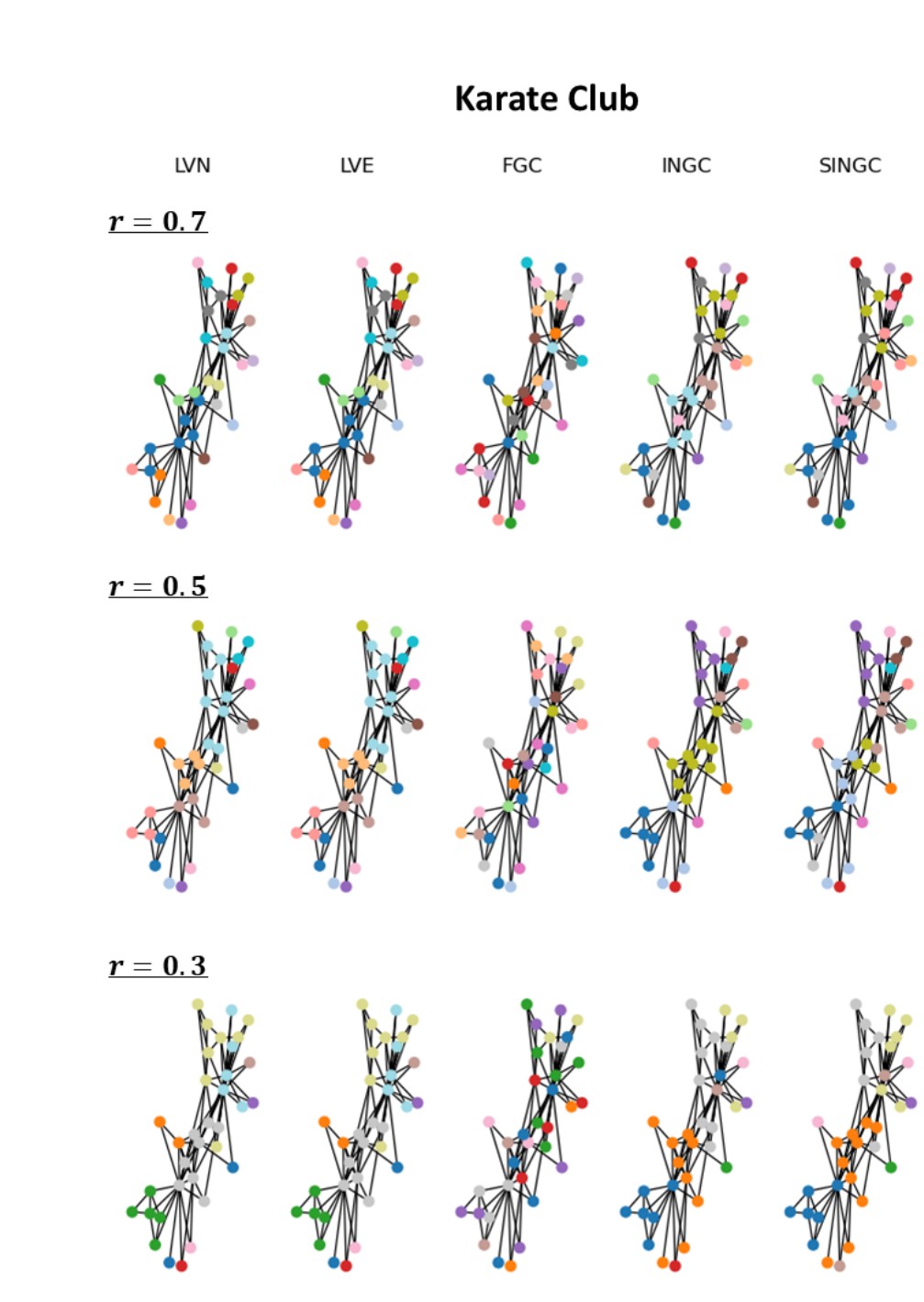

Figure 5: Visual comparison of coarsening methods on the Karate Club dataset. Each row displays the partitioning produced by each method at a different coarsening ratio. Nodes of the same color are grouped into the same super-node.

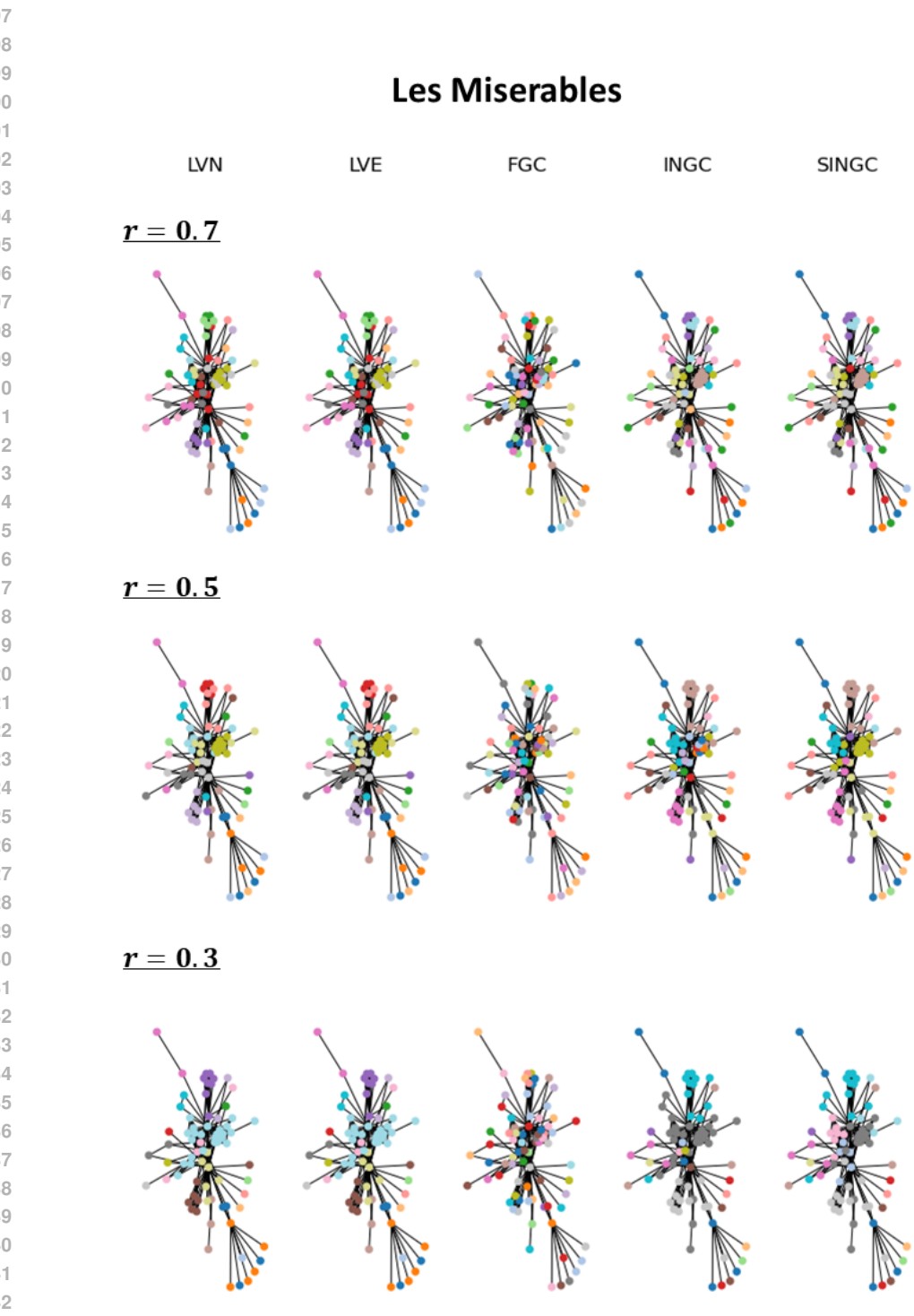

Figure 6: Visual comparison of coarsening methods on the Les Miserables dataset. Each row displays the partitioning produced by each method at a different coarsening ratio. Nodes of the same color are grouped into the same super-node.

