# OpenReview forum: "Spectral Graph Coarsening Using Inner Product Preservation and the Grassmann Manifold"
_ICLR.cc/2025/Conference — Submitted to ICLR 2025_

### Official Review · Reviewer_EEU7 · 2024-11-01

**Soundness:** 3
**Presentation:** 3
**Contribution:** 3
**Rating:** 5
**Confidence:** 4

**Summary:**

The paper proposed a method for graph coarsening that preserves the information of graph structure and node features simultaneously.

**Strengths:**

1. The idea of using IPE for graph coarsening is new.
2. The proposed method outperformed the baselines in most cases.
3. The paper also includes some theoretical results.

**Weaknesses:**

1. The implication of Theorem 3 hasn't been sufficiently explained. Particularly, is the $x_c$ in the theorem derived from the optimal solution of (12)?
2. Convergence analysis for Algorithms 1 and 2 is missing.
3. The computational complexity of the proposed algorithm hasn't been analyzed.  In addition, in Section 4.3, the authors should report the time costs of graph coarsening and GNN training (on the original graph and coarsened graph). If the time cost of coarsening is significantly higher than that of GNN training, graph coarsening is useless for accelerating GNN training.
4. The proposed algorithm has a few hyperparameters to tune but the authors haven't shown their influence and the related ablation study.

**Questions:**

1. In line 257, it was stated that the first term in (11) is not differentiable with respect to $C$. Why?
2. As the $\ell_{1,2}$ norm is nonsmooth, how did the author handle this in the optimization?
3. More explanation about the role of the second term (beta-related) in (11) should be provided.
4. What is the advantage of IPE compared to ||X^TX-Xc^TX_c|| _F^2?

---

> ### Author Response · Authors · 2024-11-24
> **Response to reviewer EEU7 - Part 1/4**
>
> Thank you for the time and effort you put into reviewing our paper. Your comments were very constructive and helped us significantly improve the manuscript. Our responses to the specific weaknesses and questions you raised and the modifications we made following them are:
>
>
> ### **Weakness 1 - Theorem 3 Implication**
>
> Theorem 3 establishes a connection between the second term in our optimization (Equation (12)) and bounds on two known coarsening metrics: REE and DEE.
> The maximum value of the second term
> $\text{tr}(U^{(k)} (U^{(k)})^\top C C^\top)$ is $k$, which occurs only when $U^{(k)}$ and $C$ represent the same point on the Grassmann manifold, i.e., $C =  O U^{(k)}$ where $O$ is an orthogonal matrix.
>
> In the theorem,  $x_c$  is the coarsened vector derived from the coarsening operator $C$ , which satisfies
> \begin{align*}
> \text{tr}(U^{(k)} (U^{(k)})^\top C C^\top) = k - \epsilon,
> \end{align*}
> where $\epsilon$ is the deviation from the optimal value of our second term. The theorem demonstrates that smaller $\epsilon$ leads to tighter bounds on REE and DEE, providing theoretical justification for including the second term in the coarsening objective.
>
> ### **Weakness 2 - Convergence Analysis**
>
> Following your comment, we added a convergence analysis in Appendix D.
>
> In Appendix D, we added a new Figure (3) that provides an illustrative example of our methods' convergence rates on two datasets. The figure demonstrates a trade-off between convergence speed and final objective loss: higher learning rates lead to faster convergence but result in a higher final loss.

---

> ### Author Response · Authors · 2024-11-24
> **Response to reviewer EEU7 - Part 2/4**
>
> ### **Weakness 3 - Complexity Analysis**
>
> Following your comment, we added a new complexity analysis section in Appendix C.
>
> The table below (also included in the new appendix) summarizes the gradient expressions and time complexities of our methods and the baseline method FGC (the only optimization-based approach among our baselines). We observe that SINGC is the most efficient, while INGC remains competitive with FGC, as both FGC and INGC are governed by  $O(n^2(k+p))$ , whereas SINGC is governed by  $O(n^2k)$ .
>
>
> |                     | FGC                                                                                                                                                                                      | INGC                                                                                                                                                                                                                                                                  | SINGC                                                                                                                                                               |
> |---------------------|------------------------------------------------------------------------------------------------------------------------------------------------------------------------------------------|-----------------------------------------------------------------------------------------------------------------------------------------------------------------------------------------------------------------------------------------------------------------------|---------------------------------------------------------------------------------------------------------------------------------------------------------------------|
> | Gradient Expression | $\nabla_{C} f(C,X_c) = 2 \big( (C X_c - X)$  $+ L (C X_c) \big) X_c^\top +\lambda C \boldsymbol{1}_{k \times k}$  $- \alpha \big( L C (C^\top L C + J)^{-1} \big)$  |  $\nabla_{C} f(C,X_c) = 2 \beta U^{(k)} ( U^{(k)} )^\top C$  $- \big[ 2 L (C X_c) \big( X^\top L X$  $- (L C X_c)^\top (C X_c) \big) X_c^\top \big]+ \lambda C \boldsymbol{1}_{k \times k}$  $- \alpha \big( L C (C^\top L C + J)^{-1} \big)$  |  $\nabla_{C} f(C) = 2 U^{(k)} ( U^{(k)} )^\top C$  $+ \lambda C \boldsymbol{1}_{k \times k}$  $- \alpha \big( L C (C^\top L C + J)^{-1} \big)$  |
> | Time Complexity     | $O\big( n^2(k + p) + k^3 \big)$                                                                                                                                           |  $O\big( n^2(k + p)+n p k + n k^2 + k^3 \big)$                                                                                                                                                                                                          |  $O\big( n^2 k + n k^2 + k^3 \big)$                                                                                                                   |
>
> **Table 1.** Comparison of gradient expressions and time complexities for FGC, INGC, and SINGC.
>
> The total time complexity for node classification on the original graph is $O(n^2lp + nle$), where $n$ is the of nodes, $e$ number of edges, $p$ number of node feature and $l$ number of layers. Since the number of coarsened nodes $k$ is typically greater than the number of node features $p$, applying coarsening before a GCN is particularly beneficial for dense graphs where $e > n$.  Coarsening reduces the graph size while keeping the dominant complexity term at $O(n^2)$.

---

> ### Author Response · Authors · 2024-11-24
> **Response to reviewer EEU7 - Part 3/4**
>
> ### **Weakness 4 -  Ablation and Hyperparameter study**
>
> In response to your comments, we have added an ablation and hyperparameter study in a new Appendix E.
>
> In appendix E, we display a new Figure (4) that presents the contribution of each parameter in our methods. It illustrates the sensitivity of each parameter and evaluates the impact of deviations from optimal values on various metrics.
>
> The figure shows that varying
> $\alpha$ results in minimal sensitivity across metrics, except for IPE, where changes up to an order of magnitude still yield similar results.
> It also demonstrates that our methods are more sensitive to
> $\lambda$ compared to the other parameters, highlighting $\lambda$'s critical role in performance.
>
> The Table below presents a new ablation study on the parameter $\beta$  - that govern the second term in our objective -  for the node classification task across various datasets and coarsening ratios $r$  (shown also in the new Appendix E). The comparison includes three methods: INGC with  $\beta = 0$  (ignoring the term  $\text{tr}(U^{(k)} (U^{(k)})^\top C C^\top)$ for minimizing IPE for general smooth signals), INGC with the optimal $\beta$ , and SINGC (our second proposed method, which omits the first term of the objective entirely).
> The table reports node classification accuracy, with the best results highlighted in bold and the second-best results underlined. For each metric, other hyperparameters are set to their optimal values. The results demonstrate the importance of balancing the two complementary approaches to minimizing IPE. INGC with  $\beta = 0$ generally underperforms compared to the other methods.
>
>
> | Dataset  |   r  |       INGC ($\beta=0$)       |             INGC            |            SINGC            |
> |----------|:----:|:----------------------------:|:---------------------------:|:---------------------------:|
> | Cora     |  0.3 |  $\underline{84.62\pm0.59}$  | $\boldsymbol{87.55\pm0.16}$ |        $84.51\pm0.33$       |
> |          |  0.1 |        $83.01\pm0.53$        | $\boldsymbol{83.38\pm0.47}$ |  $\underline{82.76\pm0.32}$ |
> |          | 0.05 |        $76.92\pm1.11$        | $\underline{77.42\pm0.78}$  | $\boldsymbol{77.81\pm0.68}$ |
> | Citeseer |  0.3 |        $76.25\pm0.28$        | $\boldsymbol{76.89\pm0.23}$ |  $\underline{76.66\pm0.27}$ |
> |          |  0.1 |        $67.07\pm0.59$        | $\boldsymbol{72.63\pm0.25}$ |  $\underline{69.71\pm0.72}$ |
> |          | 0.05 |        $60.66\pm1.58$        | $\underline{66.02\pm0.32}$  | $\boldsymbol{66.37\pm0.57}$ |
> | Pubmed   | 0.05 | $\boldsymbol{83.60\pm0.23}$  | $\boldsymbol{83.59\pm0.22}$ |  $\underline{83.55\pm0.32}$ |
> |          | 0.03 |        $81.62\pm0.14$        | $\underline{81.93\pm0.22}$  | $\boldsymbol{83.19\pm0.18}$ |
> |          | 0.01 |        $79.08\pm0.72$        | $\underline{79.09\pm0.26}$  | $\boldsymbol{79.96\pm0.34}$ |
> | Co-CS    | 0.05 |        $90.42\pm0.18$        | $\underline{90.84\pm0.12}$  | $\boldsymbol{90.92\pm0.22}$ |
> |          | 0.03 |        $89.28\pm0.21$        | $\underline{89.59\pm0.38}$  | $\boldsymbol{89.99\pm0.41}$ |
> |          | 0.01 |        $77.79\pm1.15$        | $\boldsymbol{87.93\pm0.33}$ |  $\underline{83.39\pm0.33}$ |
> | **#Best**   |      |              1              |              6              |              6              |
> | **#2-Best** |      |              1              |              6              |              5              |
>
> **Table 2.** Ablation study of the parameter  $\beta$  on node classification tasks. The table reports the accuracy on various datasets for different coarsening ratios  $r$  using different coarsening methods. The third column presents the results of our INGC method with  $\beta = 0$ , the fourth column corresponds to the optimal  $\beta$  value, and the fifth column shows the results of SINGC. Best results are in bold; second-best results are underlined. The last two rows indicate the number of times each method achieved the best and second-best performance.

---

> ### Author Response · Authors · 2024-11-24
> **Response to reviewer EEU7 - part 4/4**
>
> ### **Questions**
>
>  1. If we plug in the relation $L_c=C^T L C$ and $X_c=C^\dagger X$ to the first expression of eqaution (11) (the IPE) we get:\\
> \begin{align*}
> \|X^TLX -X^T (C^\dagger)^T C^T L C  (C^\dagger) X \|_F,
> \end{align*}
> the derivative with respect to $C$ does not have a closed-form expression.
>  2.  The $l_{1,2}$ norm used in our paper follows the same definition as in [Kumar et al. (2023)], originally defined in [Ming et al. (2023)], as $\left\lvert C^T \right\rvert_{1,2} = \sum_{i=1}^n ( \sum_{j=1}^k\left\lvert C_{i, j} \right\rvert)^2$. To handle the non-smoothness of this expression around 0, we limit the elements of $C$ to be non-negative. Then we use the relation shown in  [Kumar et al. (2023)], that this norm can be equivalently phrased as  $\left\lvert C^T \right\rvert_{1,2}  = \text{tr}(\boldsymbol{1}^\top C^\top C \boldsymbol{1})$ (see Equation (50) in their appendix), and we use the corresponding derivative.
>  3. The second term in Equation (11) minimizes the IPE for general unseen signals (node features) that satisfy Assumption 1 (smoothness on the graph). We notice that both the coarsening operator $C$ and the subspace of general smooth signals lie on the Grassmann manifold. Theorem 2 suggests that any equivalent representation of the same point on the Grassmann manifold as $U^{(k)}$ minimizes the IPE for any signal that satisfies the smoothness assumption. Therefore, the second term in our objective maximizes the geodesic similarity (defined in Equation (4)) between $C$ and $U^{(k)}$. Theorem 3 connects the optimization of this term to minimizing common graph coarsening metrics such as REE and DEE.
>  4. The expression $ X^\top X $ measures the standard inner product between two signals without considering the structure of the grid on which they are defined, i.e.,
> \begin{align*}
> \langle x, y \rangle = x^\top y = \sum_{i=1}^n x(i)y(i).
> \end{align*}
>
> Our proposed IPE incorporates the graph structure on which the signals are defined and quantifies their similarity with respect to this structure:
> \begin{align*}
> x^\top L y = \sum_{(i,j) \in E} w_{ij} (x(i) - x(j))(y(i) - y(j)),    \end{align*}
> where $w_{ij}$  are edge weights, and $x(i), y(i)$  are the values of the features at node  $i$ .
> Thus, the IPE captures both node feature and graph structure information, making it particularly beneficial for tasks where both are of interest.
> Following your question we explicitly added this relation when defining $\langle x, y \rangle_L$  and clarified its contribution throughout the paper.

---

> > ### Comment · Reviewer_EEU7 · 2024-11-26
> >
> > Thanks for the detailed response to my comments. The answer to my Q1 seems incorrect. Your statement only showed that there is no closed-form solution, which has nothing to do with the "not differentiable". For Q4, do you have experimental results to show the advantage? One additional question: How did the authors determine the three hyperparameters $\alpha,\beta,\lambda$ in the experiments?

---

> ### Author Response · Authors · 2024-11-29
> **Response to reviewer EEU7 Follow-up Questions**
>
> ### **Follow-up Questions**
>
> 1. You are correct. The derivative of this expression does not have a closed-form solution but  it may be differentiable. A key condition for $\text{pinv}(C)$ to be differentiable is that the rank of $C$ remains constant within an open neighborhood around $C$. This condition is not explicitly addressed as part of our optimization process. We thank the reviewer for this clarification and have revised our paper to reflect this distinction more accurately.
>
> 2. Following your comment, we conducted an additional experiment comparing the performance of our suggested IPE with the standard inner product:
>
> \begin{align*}
> \|X^\top X - X_c^\top X_c\|_F^2 = \|X^\top X - X^\top \text{pinv}(C)^\top \text{pinv}(C) X\|_F^2.
> \end{align*}
>
> Since the derivative of this expression with respect to  $C$  does not have a closed-form solution, we relaxed $\text{pinv}(C)$  to  $C^\top$  to avoid using derivative numerical approximations, which could result in an unfair comparison between the methods.
>
> We conducted experiments on two medium-sized datasets: Cora and Citeseer.
> The table below summarizes the performance of the two approaches across different datasets and coarsening ratios ( $r = \frac{k}{n} = 0.7, 0.5,$  and $0.3$ ). The first approach uses our proposed method (INGC) with $\beta = 0$  (i.e., using only the IPE), and the second approach that uses the standard inner product as suggested by the reviewer(SIP).
>
> | Method           |     |           |   Cora    |           |            |    Citeseer         |           | \#Best |
> |------------------|-----|:---------:|:---------:|:---------:|:---------:|:---------:|:---------:|:------:|
> |                  | **r** |   **0.7**   |   **0.5**   |   **0.3**   |   **0.7**   |   **0.5**   |   **0.3**   |        |
> |                  | REE |    0.87   |  **1.14** |  **5.18** |    0.82   |  **3.14** |  **4.39** |    4   |
> |                  | RE  |  **9.61** | **10.17** | **10.75** |    9.61   | **10.07** | **10.62** |    5   |
> | INGC ($\beta=0$) | HE  |  **0.72** |  **1.10** |  **1.67** |    0.98   |  **1.26** |  **1.94** |    5   |
> |                  | DEE |  **3e-5** |  **3e-3** |  **3e-2** |  **1e-3** |  **1e-2** |  **1e-2** |    6   |
> |                  | IPE | **32.58** | **43.67** | **68.64** | **35.52** |   44.15   | **47.79** |    5   |
> |------------------|-----|:---------:|:---------:|:---------:|:---------:|:---------:|:---------:|:------:|
> |                  | REE |  **0.86** | 3.12      | 5.41      |  **0.81** |    3.18   |    4.52   |    2   |
> |                  | RE  |    9.77   |   10.76   |   11.46   |  **9.58** | **10.07** |   10.65   |    2   |
> | SIP              | HE  |    0.85   |    1.50   |    2.27   |  **0.97** |  **1.26** |    1.98   |    2   |
> |                  | DEE |    4e-3   |    0.12   |    0.27   |  **1e-3** |  **1e-2** |    9e-2   |    2   |
> |                  | IPE |   40.16   |   75.12   |   99.54   |   39.05   | **43.16** |   55.76   |    1   |
>
> We observe that incorporating the graph structure (L) into the inner product definition is beneficial in most cases.
>
> 3. We determined the optimal hyperparameters in all our experiments through a grid search. This grid search can always be applied to optimize a specific graph coarsening metric, as evaluating the score only requires the original and coarsened graphs. We observe in Tables 7, 8, and 9 in the Appendix that the parameter values leading good performance in node classification tasks often align with low values of REE and INP. Therefore, we recommend that practitioners first optimize the hyperparameters by minimizing REE and INP, and then apply those parameters to their application.

---

> > ### Comment · Reviewer_EEU7 · 2024-12-03
> >
> > I appreciate the authors' responses to my questions and I have to keep the rating unchanged (may be changed during the final discussion period) since the paper still has the following limitations:
> > 1. Lacking of rigour. For example, previously, the authors could not distinguish between the concepts of non-differentiating and no closed-form solution. The proof for Theorem 1 is not convincing.
> > 2. The optimization algorithm is quite heuristic, without any theoretical guarantee about the convergence.
> > 3. The hyperparameter tuning remains unclear. Whether it is based on cross-validation/validation set or testing set is not clear.

---

> > > ### Author Response · Authors · 2024-12-03
> > > **Response to reviewer EEU7**
> > >
> > > We wish to clarify a few points regarding the limitations raised by the reviewer:
> > >
> > > 1.Regarding Theorem 1, we recognize that it primarily serves as a general motivation for introducing the Inner Product Error (IPE). The rigorous theoretical justification for our approach is provided by Theorems 2 and 3, which establish connections between our optimization terms and established graph coarsening metrics. We believe these theorems offer a solid foundation for our methodology. Additionally, we appreciate the reviewer's feedback regarding the distinction between non-differentiability and the lack of a closed-form solution, and we have updated the manuscript accordingly.
> > >
> > > 2.We acknowledge the concern about the lack of theoretical convergence guarantees for our optimization algorithm. Our method employs standard techniques like gradient descent and projected gradient descent, which are widely used for solving non-convex optimization problems. In Appendix D, we include plots and a brief discussion demonstrating the typical convergence behavior observed in our experiments. While theoretical convergence proofs are challenging for non-convex problems, we believe the practical performance of our algorithm validates its effectiveness.
> > >
> > > 3. We apologize for any confusion regarding hyperparameter tuning. In graph coarsening tasks, there is typically no separation between training and testing sets, as the goal is to coarsen the entire graph while preserving its properties. As such, cross-validation is not commonly used in this context. Instead, we selected the hyperparameters using a grid search aimed at minimizing specific graph coarsening metrics. This ensures that the coarsened graph retains essential characteristics of the original graph. We have clarified our hyperparameter tuning process in the revised manuscript and included an ablation and hyperparameter study in Appendix E.

---

### Official Review · Reviewer_A1KD · 2024-11-01

**Soundness:** 2
**Presentation:** 2
**Contribution:** 2
**Rating:** 5
**Confidence:** 4

**Summary:**

This paper addresses the challenge of simplifying large-scale graph data, crucial for fields such as social networks, biological systems, and recommendation systems, where graphs have become too large for traditional processing. The authors review existing graph reduction techniques: sparsification (removing edges and nodes), condensation (creating synthetic graphs for specific tasks), and coarsening (grouping similar nodes into super-nodes). While coarsening methods traditionally focus on structural properties, they often neglect node features, which are essential for many graph learning tasks. A recent approach, Featured Graph Coarsening (FGC), incorporates node features but still falls short of fully utilizing relationships between node attributes.

The authors propose a novel graph coarsening approach from a functorial perspective, treating node features as signals on the graph. Their method introduces a new metric, Inner Product Error (IPE), to measure preservation of inner product relationships between node features, aiming to maintain both structural consistency and feature relationships. This is achieved through an optimization process on the Grassmann manifold, enabling their model to generalize beyond observed features under a smoothness assumption. The method is validated through empirical results, showing that it not only maintains global structure but also outperforms state-of-the-art coarsening methods across several benchmarks, demonstrating improved utility and accuracy in graph coarsening and node classification tasks.

**Strengths:**

1. This paper introduces a functional perspective on graph coarsening by treating node features as functions (or signals) on the graph, a approach that differs significantly from traditional structural or feature-based coarsening methods.

2. It seems the authors provide a rigorous formulation of their approach, deriving a new coarsening metric and implementing a gradient descent optimization process that aligns well with the theoretical framework.

3.  By introducing IPE and optimizing on the Grassmann manifold, it seems this work potentially open doors for incorporating node feature relationships into coarsening, especially for applications like node classification that depend heavily on feature fidelity.

**Weaknesses:**

1. It is not clear why Inner Product Error (IPE), for preserving feature relationships in the coarsened graph, has the practical impact in real-world applications could be more comprehensively discussed.  For example, the authors could further clarify how preserving inner products between node features directly benefits graph-based tasks, like link prediction or graph classification.

2. The approach relies on a smoothness assumption for node features on the Grassmann manifold. However, this may limit its applicability to graphs with less smooth or heterogeneous node features. A discussion or experiment showing how the method performs under various levels of feature smoothness could clarify its robustness and potential limitations.

3. The paper's innovation is limited. It is an incremental improvement from several previous work such as presented in Loucas 2019, Kumar et al 2023, and some theoretical results are repeated from the above papers. See my questions below.

4. It seems the paper was written well overall, but some details are not clear or mistaken/wrong.  See questions below.

**Questions:**

1. The notation used in Equation (10) lacks coherence. For instance, it appears that the vector or matrix norm-0 represents the number of non-zero elements. If C_{:,i}​ denotes the i-th column of C, then by this notation, C^\top_{:,i} would be a corresponding (row) vector. Consequently, their norm-0 values should be identical. Therefore, having two conditions— one with ≥1 and another with =1— introduces a degree of inconsistency.   Please refer to Kumar et al (2023) for more exact definition

2. Theorem 1 is the repeat of Proposition 2.4 in Loucas 2019. No need to prove it again.

3. Traditional notation for matrix Frobenius norm \| M\|_F  means the square root of the sum of sqaured elements. Thus the term of Frobenous in eq (11) etc needs a square.

4. Could you please  elaboration on the second term (trace) in equation (11)?   That is your key point different from the objective used in Kumar et al (2023).  In my opinion the original objective in Kumar et al (2023) (without your first term) is more meaningful, as your first term condition is too strong.

5. Could you please given a more exact definition of l_{1,2}-norm used in the paper.  The l_{1,2} matrix norm is standard term which is defined as the sum of l_2 norms of all row (or column) vectors.  If this is the case, your derivative formula in Line 870 is incorrect.   Also I checked Kumar et al (2023) paper, I think it was defined as the sum of squared sum of rows.  Of course that was not correct too.  This way does not specify group-sparsity.  Taking sum in Kumar's case is because it was assumed for positive-element matrix. In your case, when the positive condition removed in (12) (from (10)), you need absolute operation, thus it makes your objective non-differentiable.

6. In Theorem 3, although \kappa was introduced in your proof in Appendix, but it is better to define it Theorem 3.

7. In Line 485, "we evaluate the classification performance on the original graph".  Can you give more details how this was done on the original graph?

8. You miss x in Line 783

---

> ### Author Response · Authors · 2024-11-24
> **Response to reviewer A1KD - Part 1/2**
>
> Thank you for the time and effort you put into reviewing our paper. Your comments were very constructive and helped us significantly improve the manuscript. Our responses to the specific weaknesses and questions you raised and the modifications we made following them are:
>
>
> ### **Weakness 1 - IPE Importance**
> The practical importance of the IPE lies in its ability to capture both node feature information and graph structure, making it particularly beneficial for tasks that rely on both, such as node classification and link prediction.
>
> Following your comment, we added clarifications throughout the paper, highlighting its utility. In the background section, we introduced the relation
> \begin{align*}
> x^\top L y = \sum_{(i,j) \in E} w_{ij} (x(i) - x(j))(y(i) - y(j)),    \end{align*}
>
> which provides intuition on how the inner product captures relationships between functions with respect to the graph structure.
>
> We then clarified the contribution of our theoretical guarantees. Theorem 1 explains how preserving these relationships also preserves the graph structure, while Theorem 3 demonstrates that minimizing IPE for general smooth signals ensures the preservation of important graph properties, such as dominant eigenvalues and signal norms.
>
> Finally, the motivation for using IPE is strengthened by our empirical results, where we show that it outperforms current state-of-the-art coarsening methods in common graph coarsening benchmarks and demonstrates its applicability for more efficient GNN training.
>
>
> ### **Weakness 2 - Smoothness Assumption**
>
> We wish to clarify two key points. First, please note that the signal smoothness assumption pertains to the graph structure, meaning that connected nodes tend to have similar features. Second, even when node features are not available, Theorem 2 demonstrates that maximizing the second term in our objective (which does not depend on the given node features) minimizes the IPE for any signal (including unseen ones) that satisfies the smoothness assumption. Additionally, our proposed algorithm termed SINGC does not rely on node features during the coarsening optimization.
>
> In response to your comment, we have added a clarification following the presentation of our proposed approach to better highlight the distinction between the two terms and the contributions of our work.
>
> ### **Weakness 3 - Main Contribution**
>
> We briefly review the primary contribution of our work: we propose a new graph coarsening framework that focuses on preserving the inner products of signals with respect to the graph structure. This ensures that both node feature relationships and graph structural properties are maintained during coarsening, which is crucial for downstream graph learning tasks. By recognizing that the coarsening operator and the subspace of smooth signals can both be represented as points on the Grassmann manifold, we efficiently generalize this objective to any signal satisfying a smoothness assumption, enabling us to coarsen a graph while preserving mutual information between node features, even when the node feature are unknown.
> We provide theoretical justification for our approach, link it to established coarsening metrics, and demonstrate its superior performance through extensive experiments on graph coarsening benchmarks.

---

> ### Author Response · Authors · 2024-11-24
> **Response to reviewer A1KD - Part 2/2**
>
> ### **Weakness 4 - General Questions**
>
> 1. We thank the reviewer for spotting this confusion in our notation. The first condition relates to the columns of $C$. Each column should have at least one non-zero element, denoted by
> $\left\lvert C_{\text{:}, i} \right\rvert_0 \geq 1$. The second condition relates to the rows of $C$. Every row of $C$ should have exactly one non-zero element. Following your comment, in the revised paper we changed the notation of this condition to $\left\lvert C_{i,\text{:}} \right\rvert_0= 1$ and specified after equation (10) that $C_{i,:}$  denotes the $i$-th row of C.
>
> 2.  Please note that Theorem 1 is not the same as Proposition 2.4 in Loucas 2019. Proposition 2.4 states that for any vector  $x = \Pi x$ , the norm of the signal is preserved after coarsening and lifting:
> \begin{align*}
> x_c^\top L x_c = x^\top \Pi L \Pi x = x^\top L x.
> \end{align*}
> However, it does not directly imply that the full graph structure can be reconstructed after lifting.
> In contrast, Theorem 1 states that if the inner products between all signals are preserved, then if the of the original Laplacian is less than $(n-k)$, the original Laplacian can be fully reconstructed. A key difference is a necessary condition that the rank of $L$  is less than  k (the number of super-nodes), which is not part of Proposition 2.4.
>
> 3. Thank you for this correction, in the revised paper we added a square in all relevant equations.
>
> 4. The second term in Equation (11) minimizes the IPE for general unseen signals (node features) that satisfy Assumption 1 (smoothness on the graph). Please note that both the coarsening operator
> $C$ and the subspace of general smooth signals lie on the Grassmann manifold.
> Theorem 2 suggests that any equivalent representation of the same point on the Grassmann manifold as $U^{(k)}$ minimizes the IPE for any signal that satisfies the smoothness assumption. Therefore, the second term in our objective maximizes the geodesic similarity (defined in Equation (4)) between $C$ and $U^{(k)}$. Theorem 3 connects the optimization of our second term to minimizing common graph coarsening metrics such as REE and DEE. Satisfying our first term for any two node features is a strong condition, but our second term allows this to be relaxed by focusing only on node features that satisfy a common smoothness assumption.
>
> 5. The $l_{1,2}$ norm used in our paper follows the same definition as in Kumar et al. (2023), originally defined in Ming et al. (2019), as $\left\lvert C^T \right\rvert_{1,2} = \sum_{i=1}^n ( \sum_{j=1}^k\left\lvert C_{i, j} \right\rvert)^2$. Ming et al. (2019) demonstrated that this regularization promotes sparsity within groups (the rows of $C$, in our case). Consistent with Kumar et al. (2023), we expressed this norm equivalently as $\left\lvert C^T \right\rvert_{1,2}  = \text{tr}(\boldsymbol{1}^\top C^\top C \boldsymbol{1})$  (see Equation (50) in their appendix), and our derivative formula is derived accordingly.
> You are correct that for this equivalence to hold, the elements of $C$  must be non-negative; otherwise, the derivative would also need to include $\text{sign}(C)$ . This assumption is inferred from the condition $ C \in \mathcal{C} $, where $\mathcal{C}$ is the set of valid coarsening matrices. This was unintentionally omitted from equation (12) in the original manuscript. Following your comment, we added this assumption as a constraint in the optimization problem including a full definition of the $l_{1,2}$ norm, and clarified the derivation of the derivative in the revised paper.
>
> 6. Thank you for this comment. We made sure $\kappa$ is defined at the end of the Theorem.
>
> 7. After we train the GCN on the coarsened  graph using the coarsened  Laplacian $L_c$, features matrix $X_c$, and coarsened labels $Y_c$, we apply the weights of the learned network to the full graph Laplacian and features matrix, i.e., $\hat{y}=GCN(L,X)$, and evaluate its performance based on the RMSE.
> Following your comment, we added this clarification in the revised paper.
> 8. Fixed. Thank you.

---

> > ### Comment · Reviewer_A1KD · 2024-11-26
> >
> > Thanks for authors' taking time to answer my comments and questions.   I may still feel that the notation e.g. like |C|_{1,2} is still confusing.  I will remain my score but dont objection a possible acceptance.

---

### Official Review · Reviewer_BeiK · 2024-11-03

**Soundness:** 3
**Presentation:** 3
**Contribution:** 2
**Rating:** 5
**Confidence:** 4

**Summary:**

The paper introduces a novel graph coarsening method and presents a new definition that quantifies the inner products of node features. This approach effectively preserves both the global structure of the graph and the interrelationships among node features during the coarsening process, addressing the issue in previous methods that focused on global structure while neglecting node features.

**Strengths:**

1. The proposed method in the paper not only considers the interrelationships among node features but also maintains the global structure of the graph. Additionally, it leverages the properties of the Grassmann manifold to enhance the method's generalization capabilities. Experiments on graph coarsening and node classification demonstrate the effectiveness of this approach.
2. Building on the proposed INGC, the paper also introduces a simplified version, SINGC, which improves optimization efficiency. The node classification experiments in Section 4.3 further illustrate that SINGC performs well in clustering tasks involving large datasets.
3. The derivation of the formulas in the paper is presented in a clear and engaging manner. The notation is easy to understand, and the logical flow of the derivation is coherent and well-structured, supported by ample proofs and references.

**Weaknesses:**

1. In the experimental section of Chapter 4, the comparison methods for graph coarsening are limited to the FGC(2023) and the LVN and LVE(2018). This seems insufficient to demonstrate the effectiveness of the proposed method. It would be beneficial to include comparisons with additional methods for a more comprehensive evaluation.
2. The paper lacks a complexity analysis. When introducing new definitions and solutions, it is important to provide corresponding analyses of time and space complexity.
3. The objective function (12) contains three hyperparameters: $\beta$, $\lambda$, and $\alpha$. The authors should explain how these parameters were selected and provide relevant parameter analysis experiments.
4. The effects of the last two regularization terms in equation (12), $\lambda\| \boldsymbol{C}^{T} \|_{1, 2}^{2}$ and $\alpha\operatorname{l o g}{d e t} ( \boldsymbol{L}_{c}+\boldsymbol{J} )$, on the overall process are unclear, as there is a lack of relevant ablation experiments.

**Questions:**

1. In Table 1 of Section 4.2, titled "GRAPH COARSENING METRICS," there is a metric labeled "INP," but the definition of this metric does not appear to be mentioned elsewhere in the text. Could you provide the specific mathematical expression for it?
2. Definition 5 states that the motivation for the Inner Product Error (IPE) is based on Theorem 1, which requires the graph to have (n−k) connected components. Do the datasets used in the experiments satisfy this condition? If the method is applied to other datasets, must they also meet this condition? If a dataset does not satisfy this requirement (i.e., if the graph has more or fewer than (n−k) connected components), how would that affect the experimental results?
3. We noticed that the coarsening rates chosen for the graph coarsening experiments (Table 1) and the node classification experiments (Table 2) differ. Should different experiments with various datasets have carefully selected coarsening rates? Would it be possible to conduct experiments with a uniform coarsening rate in the range of (0.3, 0.5, 0.7)?

---

> ### Author Response · Authors · 2024-11-24
> **Response to reviewer BeiK - Part 1/3**
>
> Thank you for the time and effort you put into reviewing our paper. Your comments were very constructive and helped us significantly improve the manuscript. Our responses to the specific weaknesses and questions you raised and the modifications we made following them are:
>
> ### **Weakness 1 -  Comparison to Other Methods**
>
> In our experimental setting, we compared our results to methods considered state-of-the-art in their respective contexts. LVN and LVE are known to perform best for evaluating graph coarsening metrics that measure how well graph structural properties are preserved (e.g., REE and RE). FGC is widely regarded as the leading method for metrics that also consider node features, as it incorporates node features into the coarsening process. Thus, Section 4.2 focused on comparing our performance against these methods across various graph metrics.
>
> In Section 4.3, we repeated the experimental setting used in FGC (2023) and SCAL (2021) and reported the results of only the top-performing method in each of these settings from those papers. This means that our method also outperforms the other multigrid coarsening approaches, such as those proposed by Livne et al. (2012) [1] and Ron et al. (2011) [2], as reported in SCAL (2021).
>
> [1] Oren E Livne and Achi Brandt. Lean algebraic multigrid (lamg): Fast graph laplacian linear solver. SIAM
> Journal on Scientific Computing, 34(4):B499–B522, 2012.
>
> [2] Dorit Ron, Ilya Safro, and Achi Brandt. Relaxation-based coarsening and multiscale graph organization.
> Multiscale Modeling and Simulation, 9(1):407–423, 2011.
>
> ### **Weakness 2 -  Time Complexity**
>
> Following your comment, we added a new complexity analysis section in Appendix C.
>
> The table below (also included in the new appendix) summarizes the gradient expressions and time complexities of our methods and the baseline method FGC (the only optimization-based approach among our baselines). We observe that SINGC is the most efficient, while INGC remains competitive with FGC, as both FGC and INGC are governed by  $O(n^2(k+p))$ , whereas SINGC is governed by  $O(n^2k)$ .
>
>
> |                     | FGC                                                                                                                                                                                      | INGC                                                                                                                                                                                                                                                                  | SINGC                                                                                                                                                               |
> |---------------------|------------------------------------------------------------------------------------------------------------------------------------------------------------------------------------------|-----------------------------------------------------------------------------------------------------------------------------------------------------------------------------------------------------------------------------------------------------------------------|---------------------------------------------------------------------------------------------------------------------------------------------------------------------|
> | Gradient Expression | $\nabla_{C} f(C,X_c) = 2 \big( (C X_c - X)$  $+ L (C X_c) \big) X_c^\top +\lambda C \boldsymbol{1}_{k \times k}$  $- \alpha \big( L C (C^\top L C + J)^{-1} \big)$  |  $\nabla_{C} f(C,X_c) = 2 \beta U^{(k)} ( U^{(k)} )^\top C$  $- \big[ 2 L (C X_c) \big( X^\top L X$  $- (L C X_c)^\top (C X_c) \big) X_c^\top \big]+ \lambda C \boldsymbol{1}_{k \times k}$  $- \alpha \big( L C (C^\top L C + J)^{-1} \big)$  |  $\nabla_{C} f(C) = 2 U^{(k)} ( U^{(k)} )^\top C$  $+ \lambda C \boldsymbol{1}_{k \times k}$  $- \alpha \big( L C (C^\top L C + J)^{-1} \big)$  |
> | Time Complexity     | $O\big( n^2(k + p) + k^3 \big)$                                                                                                                                           |  $O\big( n^2(k + p)+n p k + n k^2 + k^3 \big)$                                                                                                                                                                                                          |  $O\big( n^2 k + n k^2 + k^3 \big)$                                                                                                                   |
>
> **Table 1.** Comparison of gradient expressions and time complexities for FGC, INGC, and SINGC.

---

> ### Author Response · Authors · 2024-11-24
> **Response to reviewer BeiK - Part 2/3**
>
> ### **Weakness 3 and 4 -  Ablation and Hyperparameter study**
>
> We selected the hyperparameters in our experiments through a grid search. In response to your comments, we have added an ablation and hyperparameter study in a new Appendix E.
>
> In appendix E, we display a new Figure (4) that presents the contribution of each parameter in our methods. It illustrates the sensitivity of each parameter and evaluates the impact of deviations from optimal values on various metrics.
>
> The figure shows that varying
> $\alpha$ results in minimal sensitivity across metrics, except for IPE, where changes up to an order of magnitude still yield similar results.
> It also demonstrates that our methods are more sensitive to
> $\lambda$ compared to the other parameters, highlighting $\lambda$'s critical role in performance.
>
> The Table below presents a new ablation study on the parameter $\beta$  - that govern the second term in our objective -  for the node classification task across various datasets and coarsening ratios $r$  (shown also in the new Appendix E). The comparison includes three methods: INGC with  $\beta = 0$  (ignoring the term  $\text{tr}(U^{(k)} (U^{(k)})^\top C C^\top)$ for minimizing IPE for general smooth signals), INGC with the optimal $\beta$ , and SINGC (our second proposed method, which omits the first term of the objective entirely).
> The table reports node classification accuracy, with the best results highlighted in bold and the second-best results underlined. For each metric, other hyperparameters are set to their optimal values. The results demonstrate the importance of balancing the two complementary approaches to minimizing IPE. INGC with  $\beta = 0$ generally underperforms compared to the other methods.
>
>
> | Dataset  |   r  |       INGC ($\beta=0$)       |             INGC            |            SINGC            |
> |----------|:----:|:----------------------------:|:---------------------------:|:---------------------------:|
> | Cora     |  0.3 |  $\underline{84.62\pm0.59}$  | $\boldsymbol{87.55\pm0.16}$ |        $84.51\pm0.33$       |
> |          |  0.1 |        $83.01\pm0.53$        | $\boldsymbol{83.38\pm0.47}$ |  $\underline{82.76\pm0.32}$ |
> |          | 0.05 |        $76.92\pm1.11$        | $\underline{77.42\pm0.78}$  | $\boldsymbol{77.81\pm0.68}$ |
> | Citeseer |  0.3 |        $76.25\pm0.28$        | $\boldsymbol{76.89\pm0.23}$ |  $\underline{76.66\pm0.27}$ |
> |          |  0.1 |        $67.07\pm0.59$        | $\boldsymbol{72.63\pm0.25}$ |  $\underline{69.71\pm0.72}$ |
> |          | 0.05 |        $60.66\pm1.58$        | $\underline{66.02\pm0.32}$  | $\boldsymbol{66.37\pm0.57}$ |
> | Pubmed   | 0.05 | $\boldsymbol{83.60\pm0.23}$  | $\boldsymbol{83.59\pm0.22}$ |  $\underline{83.55\pm0.32}$ |
> |          | 0.03 |        $81.62\pm0.14$        | $\underline{81.93\pm0.22}$  | $\boldsymbol{83.19\pm0.18}$ |
> |          | 0.01 |        $79.08\pm0.72$        | $\underline{79.09\pm0.26}$  | $\boldsymbol{79.96\pm0.34}$ |
> | Co-CS    | 0.05 |        $90.42\pm0.18$        | $\underline{90.84\pm0.12}$  | $\boldsymbol{90.92\pm0.22}$ |
> |          | 0.03 |        $89.28\pm0.21$        | $\underline{89.59\pm0.38}$  | $\boldsymbol{89.99\pm0.41}$ |
> |          | 0.01 |        $77.79\pm1.15$        | $\boldsymbol{87.93\pm0.33}$ |  $\underline{83.39\pm0.33}$ |
> | **#Best**   |      |              1              |              6              |              6              |
> | **#2-Best** |      |              1              |              6              |              5              |
>
> **Table 2.** Ablation study of the parameter  $\beta$  on node classification tasks. The table reports the accuracy on various datasets for different coarsening ratios  $r$  using different coarsening methods. The third column presents the results of our INGC method with  $\beta = 0$ , the fourth column corresponds to the optimal  $\beta$  value, and the fifth column shows the results of SINGC. Best results are in bold; second-best results are underlined. The last two rows indicate the number of times each method achieved the best and second-best performance.

---

> ### Author Response · Authors · 2024-11-24
> **Response to reviewer BeiK - Part 3/3**
>
> ### **Questions**
>
> 1. This is a typo. We meant IPE (as defined in definition 5) and fixed it. Thank you.
> 2. The assumption in Theorem 1 aids in mathematical tractability and rigorous derivation. In practice, most graphs are connected or have only a few components. However, we show that even when this criterion is not met (as in all of our datasets), our method still achieves the lowest reconstruction error (RE) compared to other methods, highlighting its broader applicability.
> 3. We wish to clarify our choice of coarsening rates. In Table 2, small coarsening ratios were used to support effective downscale for GNNs, and the specific ratio values were chosen to align with the baseline experimental setups for fair comparison. In Table 1, which involves small and medium-sized datasets, using the same coarsening ratios as in Table 2 (e.g., 0.1 or 0.05) would result in overly small coarsened graphs, losing meaningful structure.

---

### Official Review · Reviewer_4R8Q · 2024-11-04

**Soundness:** 3
**Presentation:** 3
**Contribution:** 3
**Rating:** 8
**Confidence:** 4

**Summary:**

The paper proposes a novel graph coarsening strategy for graph neural networks based on the ability of the polling to preserve the input features and in general smooth features defined on the nodes of the original graph. In particular, the coarsening is performed through a coarsening matrix C (over the Stiefel manifold?), which is optimized to minimize |X^T L X - X_C^T L_C X_C|, where X_C and L_C are the coarsened features and laplacian matrices. Moreover, a further additional term promotes the preservation of smooth functions by minimizing the distance of C from the first k smallest eigenvalues of the Laplacian on the Grassmann manifold. Two variants are tested (with and without feature preservation loss) and compared on general coarsening metrics and node classification tasks.

**Strengths:**

The paper is generally well-written and easy to follow. My only comment is about introducing the importance given to the Grassmann manifold in the background and the relatively low importance given to the second term of the loss using it. At first, I was confused, not understanding where the Grassman manifold was coming into play in the definition of the first loss, to which most of section 3 (at least 3 and 3.1) is dedicated.

The proposed methodology is sound and theoretically founded (except for the first term, see weaknesses). The authors made theoretical connections between the proposed loss and some of the metrics used for evaluating graph coarsening methods.

The method compares favorably with other coarsening methods in most datasets and metrics.

**Weaknesses:**

* I’m not sure eq 6 can be seen as the dot product between signals over nodes. Do you have any references for this?  For instance,” x^T L y” would be zero for any constant value of x and y. It might be interpreted as capturing some relationship between the smoothness of x and y, but I’m not sure.

* Coarsening is posed as an optimization problem. This might be a problem on larger graphs, making the whole point of graph coarsening methods fail. It would be nice to understand what graph size the method can work with and the convergence speed/time of the methods compared with others.

**Questions:**

Considering my previous comment on eq 6, I would like to understand how important it is to consider the relation between different functions rather than just the trace of eq6. In this case, wouldn’t your formulation contrast with FGC?

Minor:
* Fix the bibliography by updating arXiv with the published version when it exists.
* is the definition of L_c missing in 12
* what is mathcal{C} in eq 11?
* ordering of terms is not consistent between appendix and main (e.g. eq 19 and 20)

---

> ### Author Response · Authors · 2024-11-24
> **Response to reviewer 4R8Q - Part 1/2**
>
> Thank you for the time and effort you put into reviewing our paper. Your comments were very constructive and helped us significantly improve the manuscript. Our responses to the specific weaknesses and questions you raised and the modifications we made following them are:
>
> ### **Weakness 1 - Dot Product Definition**
>
> You are correct that  $x^\top L x = 0$  for any constant vector  $x$ , and thus  $x^\top L x$  does not induce a norm but rather a semi-norm. In response to your comment, we have added a clarification in the revised paper that  $x^\top L y$  can be viewed as an inner product on the subspace of  $\mathbb{R}^n$  orthogonal to the constant vector  $\mathbf{1}$ , following several prior works on spectral graph theory (e.g., [Von Luxburg, 2007]).
>
>
> Additionally, we clarified that the inner product indeed captures signal smoothness wrt the graph by adding the explicit expression:
>
> \begin{align*}
> x^\top L y = \sum_{(i,j) \in E} w_{ij} (x(i) - x(j))(y(i) - y(j)),
> \end{align*}
>
> where $ w_{ij} $ are edge weights and $x(i), y(i)$  are the values of the signals at node  $i$ . This form measures the variation and alignment of  $x$  and  $y$  across connected nodes, reflecting their relationship with the graph structure.
>
> ### **Weakness 2a - Complexity Analysis**
>
> Following your comment, we added a new complexity analysis section in Appendix C.
>
> The table below (also included in the new appendix) summarizes the gradient expressions and time complexities of our methods and the baseline method FGC (the only optimization-based approach among our baselines). We observe that SINGC is the most efficient, while INGC remains competitive with FGC, as both FGC and INGC are governed by  $O(n^2(k+p))$ , whereas SINGC is governed by  $O(n^2k)$ .
>
>
> |                     | FGC                                                                                                                                                                                      | INGC                                                                                                                                                                                                                                                                  | SINGC                                                                                                                                                               |
> |---------------------|------------------------------------------------------------------------------------------------------------------------------------------------------------------------------------------|-----------------------------------------------------------------------------------------------------------------------------------------------------------------------------------------------------------------------------------------------------------------------|---------------------------------------------------------------------------------------------------------------------------------------------------------------------|
> | Gradient Expression | $\nabla_{C} f(C,X_c) = 2 \big( (C X_c - X)$  $+ L (C X_c) \big) X_c^\top +\lambda C \boldsymbol{1}_{k \times k}$  $- \alpha \big( L C (C^\top L C + J)^{-1} \big)$  |  $\nabla_{C} f(C,X_c) = 2 \beta U^{(k)} ( U^{(k)} )^\top C$  $- \big[ 2 L (C X_c) \big( X^\top L X$  $- (L C X_c)^\top (C X_c) \big) X_c^\top \big]+ \lambda C \boldsymbol{1}_{k \times k}$  $- \alpha \big( L C (C^\top L C + J)^{-1} \big)$  |  $\nabla_{C} f(C) = 2 U^{(k)} ( U^{(k)} )^\top C$  $+ \lambda C \boldsymbol{1}_{k \times k}$  $- \alpha \big( L C (C^\top L C + J)^{-1} \big)$  |
> | Time Complexity     | $O\big( n^2(k + p) + k^3 \big)$                                                                                                                                           |  $O\big( n^2(k + p)+n p k + n k^2 + k^3 \big)$                                                                                                                                                                                                          |  $O\big( n^2 k + n k^2 + k^3 \big)$                                                                                                                   |
>
> **Table 1.** Comparison of gradient expressions and time complexities for FGC, INGC, and SINGC.
>
> The total time complexity for node classification on the original graph is $O(n^2lp + nle$), where $n$ is the of nodes, $e$ number of edges, $p$ number of node feature and $l$ number of layers. Since the number of coarsened nodes $k$ is typically greater than the number of node features $p$, applying coarsening before a GCN is particularly beneficial for dense graphs where $e > n$.  Coarsening reduces the graph size while keeping the dominant complexity term at $O(n^2)$.

---

> ### Author Response · Authors · 2024-11-24
> **Response to reviewer 4R8Q - Part 2/2**
>
> ### **Weakness 2b - Convergence Analysis**
> Following your comment, we added a convergence analysis in Appendix D.
>
> In Appendix D, we added a new Figure (3) that provides an illustrative example of our methods' convergence rates on two datasets. The figure demonstrates a trade-off between convergence speed and final objective loss: higher learning rates lead to faster convergence but result in a higher final loss.
>
>
> ### **Questions:**
>
> 1. The trace $ \text{tr}(X^\top L X) $ measures the smoothness of individual signals with respect to the graph but neglects the relationships between different signals. Our approach considers the full term $ \|X^\top L X\|_F $, which incorporates both the smoothness of individual signals and their relationships with respect to the graph structure. This ensures the preservation of both node-level information and critical structural properties, as validated by our empirical results and theoretical analysis.
>
>    Our approach can be considered a generalization of FGC, as it focuses not only on signal norms ($ \langle x, x \rangle_L $) but also on the cross-relations.
>
> 2. Fixed. Thank you.
>
> 3. Fixed. Thank you.
>
> 4. $ \mathcal{C} $ is the group of valid coarsening matrices as presented in equation (10). Following this comment, we added a clarification after presenting equation (11) in the updated manuscript.
>
> 5. Fixed. Thank you.

---

> > ### Comment · Reviewer_4R8Q · 2024-11-25
> >
> > I thank the authors for answering my questions. I will keep my positive score.

---

### Author Response · Authors · 2024-12-01
**Response to Reviewers**

We thank all the reviewers for their valuable feedback, which helped us improve our paper. We hope we have addressed all your concerns in the revised paper and in the detailed comments below.

---

### Meta-Review · Area_Chair_2sH2 · 2024-12-20

**Metareview:**

Thanks for your submission to ICLR.

The reviews were somewhat mixed on this paper, with one positive score and three leaning towards reject.  There were several issues raised by the initial reviews, including a lack of convergence analysis, some missing experiments/baselines, missing complexity analysis, missing details on various aspects of the algorithm, and limited novelty.  The authors responded to these issues, but during the discussion, some of the reviewers still did not feel that the paper is ready for publication.  There were still some lingering issues (for example, the reviewers were still concerned about a lack of convergence analysis, and about a general lack of rigor in the paper).

It seems that there are still enough issues left that remain unresolved that this paper would benefit from an additional round of editing and review.  I would encourage the authors to keep working on the paper, and to keep in mind the suggestions of the reviewers when preparing a future version of the manuscript.

**Additional Comments On Reviewer Discussion:**

As noted above, there were several points raised during the initial reviews.  Some of these were resolved in the discussion; others were not.  In particular, reviewers remained concerned about the lack of rigor and lack of convergence analysis.  These helped clarify that the paper is not ready for publication at this time.

---

### Decision · Program_Chairs · 2025-01-22

Reject